# Natural Counterfactuals With Necessary Backtracking

## Abstract

Counterfactual reasoning is pivotal in human cognition and especially important for providing explanations and making decisions. While Judea Pearl's influential approach is theoretically elegant, its generation of a counterfactual scenario often requires interventions that are too detached from the real scenarios to be feasible. In response, we propose a framework of *natural counterfactuals* and a method for generating counterfactuals that are more grounded in empirical evidence. Our methodology relaxes the non-backtracking requirement in standard counterfactual reasoning and allows changes in variables that are causally prior to the variables involved in a counterfactual supposition, when such changes are needed to satisfy some "naturalness" criterion. On the other hand, we introduce an optimization framework to encourage satisfying naturalness with as little backtracking as possible. As we show in experiments, our approach is better at generating desired counterfactual instances than the standard Pearlian approach.

## 1 Introduction

Counterfactual reasoning, which aims to answer what would have been the case if certain features of the actual situation had been different, is often used in human cognition, underpinning our ability to simulate actions, perform self-reflection, provide explanations, and inform decisions. For AI systems to mirror such human-like decision-making processes, incorporating counterfactual reasoning is crucial. Judea Pearl's structural approach to counterfactual modeling and reasoning stands as a cornerstone in machine learning. Within this framework, counterfactuals are conceptualized as being generated by surgical interventions on $X$ that sever causal links while leaving its causally upstream variables untouched (Pearl, 2009). Such non-backtracking counterfactual reasoning (i.e., directly performing an intervention on $X$ and leaving $X$'s causal ancestors intact) can yield valuable insights into the consequences of hypothetical actions. Consider a scenario: Tom on a high-speed bus fell after a sudden break and as a result injured Jerry. The non-backtracking counterfactual reasoning would tell us that if Tom had stood still (despite the sudden braking), then Jerry would not have been injured. Pearl's approach supplies a principled machinery to reason about conditionals of this sort, which are usually useful for explanation, planning, and responsibility allocation.

However, such surgical interventions may not be feasible in practice and then may not help too much in self-reflection. In this example, preventing Tom's fall in a sudden braking scenario contradicts mechanisms that may be extremely hard or even impossible to disrupt, like the inertia laws of physics. This incongruity poses challenges to the practical application of such counterfactual reasoning. From a legal perspective, Tom's fall to cause Jerry's injury could be given a "necessity defense", noting that the sudden braking has left him with no alternatives (Conde, 1981). Thus, the non-backtracking counterfactual about Tom standing still despite the sudden braking may be of very limited relevance to practical concerns in such cases.

We consider the scenario where a primary goal of counterfactual reasoning is constructive — to reflect upon and better situations. With this objective in mind, it is expected to provide actionable insights. In view of this challenge to non-backtracking counterfactuals, we introduce "natural counterfactuals" in this paper, which are counterfactuals derived from interventions that are natural, in the sense of being sufficiently plausible within observed data contexts. In the above example, it is too implausible for Tom not to fall given a sudden stop. A more plausible intervention might be one targeting a prior event, such as making the bus slow down earlier, which amounts to some backtracking.

Natural counterfactuals are therefore expected to align better with real-world scenarios and facilitate more practical solutions or more realistic decisions. In machine learning, while non-backtracking counterfactuals can sometimes produce unlikely or unrealistic scenarios, our methodology relies on empirically supported interventions, and hence mitigate the risk of implausible outcomes and ensure empirically grounded results.

In the standard Pearlian framework, counterfactual reasoning is concerned with inferences of the consequences of a change that is due to an intervention that directly brings about the change. In our approach, however, such direct interventions will be avoided if they are not sufficiently "natural", and interventions on causally preceding variables will be invoked when necessary. Our natural counterfactuals are intended to stay closer to reality than the standard Pearlian counterfactuals, by locating changes that are feasible and effective at realizing the desired counterfactual supposition. Our approach significantly enhances the real-world applicability of counterfactual reasoning, as evidenced by experiments showing an improved quality of counterfactual generation. More specifically, this paper makes the following contributions:

- We develop a general framework of what we call natural counterfactuals, which are both more flexible and more realistic than the standard framework.
- We propose a novel optimization framework for generating natural counterfactuals. By combining a naturalness constraint with a principle of minimal change that discourages unnecessary backtracking, we seek to strike a best balance between natural and non-backtracking counterfactuals.
- We present a detailed method in the general framework and test it empirically. The empirical results, on both simulated and real data, demonstrate the efficacy of our method.

## 2 RELATED WORK

**Non-backtracking Counterfactual Generation.** As will become clear, our theory is presented in the form of counterfactual sampling or generation. (Pawlowski et al., 2020b; Kocaoglu et al.; Dash et al., 2022; Sanchez & Tsaftaris) use the deep generative models to learn an SCM from data given a causal graph; these works strictly follow Pearl's theory of non-backtracking counterfactuals. Our case studies will examine some of these models and demonstrate their difficulties in dealing with unseen inputs.

**Backtracking Counterfactuals.** Backtracking in counterfactual reasoning has drawn plenty of attention in philosophy (Hiddleston, 2005; Khoo, 2017), psychology(Dehghani et al., 2012), and cognitive science (Gerstenberg et al., 2013). (Hiddleston, 2005) proposes a theory that is in spirit similar to ours, in which backtracking is allowed but limited by some requirement of matching as much causal upstream as possible. Gerstenberg et al. (2013) shows that people use both backtracking and non-backtracking counterfactuals in practice and tend to use backtracking counterfactuals when explicitly required to explain causes for the supposed change in a counterfactual. von Kügelgen et al. (2022) is a most recent paper explicitly on backtracking counterfactuals. The main differences between that work and ours are that von Kügelgen et al. (2022) requires backtracking all the way back to exogenous noises and measures closeness on noise terms, which in our view are less reasonable than limiting backtracking to what we call "necessary backtracking" and measuring closeness directly on endogenous, observable variables. Moreover, their backtracking counterfactuals sometimes allow gratuitous changes, as we explain in the Appendix.

**Algorithmic Recourse.** Algorithmic recourse emphasizes providing individuals with actionable recommendations to achieve a desired outcome from a predictive model (Karimi et al., 2020a; Upadhyay et al., 2021; Karimi et al., 2020b; Rawal et al., 2020; Pawelczyk et al., 2022). The concepts of (non-)backtracking counterfactuals and algorithmic recourse both revolve around the idea of understanding and potentially altering decisions made by predictive models. However, they target different aspects and objectives of the interpretability and fairness spectrum in AI. The primary focus here is on guiding individuals on what they can change to achieve a desired outcome from a model. Though natural counterfactuals might offer suggestions to individuals on potential interventions, they operate within a broader framework, focusing on the feasibility of these interventions rather than their direct applicability to individual cases.

**Counterfactual Explanations.** A prominent approach in explainable AI is counterfactual explanation (Wachter et al., 2018; Dhurandhar et al., 2018; Mothilal et al., 2020; Barocas et al., 2020; Pawlowski et al., 2020a; Verma et al., 2020; Schut et al., 2021; Karimi et al., 2020a), on which our work is likely to have interesting bearings. Most works on this topic define some sense of minimal changes of an input sample with a predicted class $c$ such that adding the minimal changes into the input would make it be classified into another (more desirable) class. Although this paper does not discuss counterfactual explanations, our framework may well be used to define a novel notion of counterfactual explanation by requiring the counterfactual instances to be "natural" in our sense.

## 3 NOTATIONS AND BACKGROUND

We use a **structural causal model (SCM)** to represent the data generating process of a causal system. A SCM is a mathematical structure consisting of a triplet $\mathcal{M} := < \mathbf{U}, \mathbf{V}, \mathbf{F} >$, with two disjoint variable sets $\mathbf{U} = \{\mathbf{U}_1, ..., \mathbf{U}_N\}$ (exogenous or noise variables whose values or probability distributions are given as inputs) and $\mathbf{V} = \{\mathbf{V}_1, ..., \mathbf{V}_N\}$ (endogenous or observed variables whose values are determined by other variables in the model), and a function set $\mathbf{F} = \{f_1, ..., f_N\}$, one for each endogenous variable. Note that we assume there is an equal number of exogenous and endogenous variables, or in other words, there is exactly one distinct noise variable affecting each endogenous variable. This assumption can be relaxed but we will adopt it in this paper for simplicity. Each function, $f_i \in \mathbf{F}$, specifies how an endogenous variable $\mathbf{V}_i$ is determined by its direct causes, $\mathbf{PA}_i \subseteq \mathbf{V}$:

$$\mathbf{V}_i := f_i(\mathbf{PA}_i, \mathbf{U}_i), \quad i = 1, ..., N \tag{1}$$

We assume the SCM is recursive, meaning that the transitive closure of the direct causal relations will not relate any variable to itself. A **probabilistic SCM** is a pair, $< \mathcal{M}, p(\mathbf{U}) >$, consisting of a (recursive) SCM and a probability distribution $p(\mathbf{U})$ for the noise variables. Since the SCM is recursive, $p(\mathbf{U})$ induces a joint distribution over $\mathbf{U}$ and $\mathbf{V}$. A **causal world** is a pair $< \mathcal{M}, \mathbf{u} >$, with $\mathbf{u}$ a specific value setting of $\mathbf{U}$.

A counterfactual ponders what would happen in a scenario that differs from the actual one in a certain way. Following a standard notation, terms with with a $*$ superscript refer to a counterfactual world. For example, $\mathbf{u}_k^*$ denotes $\mathbf{U}_k$'s value in a counterfactual world. Let $\mathbf{A}, \mathbf{B}, \mathbf{E}$ be sets or vectors of endogenous variables. Here is a general counterfactual question: given the observation of $\mathbf{E} = \mathbf{e}$, what would the value $\mathbf{B}$ have been if the value of $\mathbf{A}$ were $\mathbf{a}^*$ (instead of the actual value $\mathbf{a}$)? The Pearlian, non-backtracking reading of this question takes the counterfactual supposition of $\mathbf{A} = \mathbf{a}^*$ to be realized by an intervention on $\mathbf{A}$ (which will leave every causal ancestor of $\mathbf{A}$ in the model invariant). Given this understanding, counterfactual inference includes three steps. (1) Abduction: The noise distribution is updated based on the given evidence $\mathbf{E} = \mathbf{e}$. (2) Action: The causal model is modified, in which $\mathbf{A}$ is fixed to $\mathbf{a}^*$ while keeping other components the same as before. (3) Prediction: The counterfactual outcome of $\mathbf{B}$, denoted as $\mathbf{B}^*$, is inferred using the modified model and the updated noise distribution. In this setting, the counterfactual world is stipulated to share the same values of the noise variables with the actual world, and breaks by intervention some causal links that are present in the actual world. However, some interventions, e.g., having Tom stand still while keeping the sudden brake, might be infeasible in reality.

## 4 A FRAMEWORK OF NATURAL COUNTERFACTUALS

### 4.1 OVERVIEW

We aim to address this issue by relaxing the stipulation of no backtracking on the one hand and minimizing the extent of backtracking on the other. The general idea is to use a "naturalness" criterion to detect the need of backtracking and use a novel conception of minimal change to determine a minimal extent of backtracking that is needed. Multiple criteria of naturalness may be explored in this context, and we will consider some in Sec. 4.2. This discussion will be followed by a theoretical discussion of minimal change in Sec. 4.3. Sec. 4.4 will unfold a unified optimization framework to realize natural counterfactuals by integrating naturalness and minimal change.

### 4.2 NATURALNESS CRITERIA

We will refer to $p(\mathbf{V}_i|\mathbf{PA}_i)$, $i = 1, ..., N$, as *local (observed) mechanisms*, for it is in principle estimable from the observed data and the given causal structure. Note that a local mechanism also implicitly encodes the properties of noise variables; given a fixed value for $\mathbf{PA}_i$, noise $\mathbf{U}_i$ entirely dictates the probability of $\mathbf{V}_i$. Throughout this paper, the term "local mechanism" will be used to encompass both the conditional distribution of an endogenous variable given its causal parents and the distribution of its corresponding noise variable.

In our conception of natural counterfactuals, to say that $\mathbf{A} = \mathbf{a}^*$ is a feasible counterfactual supposition is to say that it could have occurred in a way that conforms to the local mechanisms in the actual world. In other ways, we need to assess whether any local mechanism that contributes to the realization of $\mathbf{A} = \mathbf{a}^*$ has seen a significant departure from that in the actual world. Let $G(\mathbf{A})$ denote the $\mathbf{A}$-realization set, which includes $\mathbf{A}$ and all its endogenous ancestors in the given SEM. Our task is to monitor the plausibility of each local mechanism within $G(\mathbf{A})$.

A minimal standard in this vein is to require that each local mechanism $p(\mathbf{N}|\mathbf{PA}(\mathbf{N}))$ taking a specific value $\mathbf{N} = \mathbf{n}$ be greater than 0, where $\forall \mathbf{N}, \mathbf{N} \in G(\mathbf{A})$. However, this standard still allows counterfactual scenarios that are too far-fetched, even though not outright impossible, which might still be problematic in practical applications. For instance, in normal distributions, nearly all observed data fall within three standard deviations from the mean. Data points outside this range (beyond the 3-sigma area) are deemed as improbable or outlier events. To steer clear of these improbable data points, we advocate more stringent naturalness criteria, with a user-adjustable threshold that can be tailored to suit specific problems at hand. [1]

#### 4.2.1 NATURALNESS MEASUREMENT

Many criteria may be proposed to capture our intention in one way or another. Below are some examples that strike us as sensible:

**Definition 1 (Measurements of Naturalness)** *Given a data point $X = x$, $\mathbf{U}_x = \mathbf{u}_X$, its parents taking a value, i.e., $\mathbf{PA}(X) = \mathbf{pa}_X$, and its local mechanism $p(X|\mathbf{PA}(X))$, four candidate measurements of naturalness are:*

*(1) $p(x|\mathbf{pa}_X)e^{H(X|\mathbf{pa}_X)}$, where $H(X|\mathbf{pa}_X) = \mathbb{E}[p(X|\mathbf{pa}_X)]$: Here, the average naturalness of variable $X$ is normalized to be 1 relative to $p(X|\mathbf{pa}_X)$. Specifically, when $\log(p(x|\mathbf{pa}_X)) = [-H(X|\mathbf{pa}_X)]$, the naturalness of such a point $x$ is 1.*

*(2) $\frac{p(x|\mathbf{pa}_X))}{\mathbb{E}[p(X|\mathbf{pa}_X)]}$: This metric aligns with (1) in setting the average naturalness to 1. Interestingly, the metric servers the low bound of (1).*

*(3) $P(u_X)$: This represents the the cumulative distribution function (CDF) of the noise value $u_X$ associated with $X$: when the value of $X$'s parents are fixed, $P(u_X)$ serves as an indicator of naturalness, where $P$ is CDF of $U_X$, i.e., $P(u_x) = \int_{-\infty}^{u_x} p(U_X)dU_X$. The closer $P(u_X)$ is to $0.5$ (meaning it is farther from the distribution tails), the higher the naturalness of the local mechanism.*

*(4) $P(x|\mathbf{pa}_X)$: the CDF value of $x$ given $\mathbf{pa}_X$: This definition is similar to (4), but it uses the cumulative distribution function $P(X)$ instead. Again, the further it is away from tails, the greater the naturalness of the local mechanism is.*

Definition (1) provides a measure of naturalness based on the concept of entropy. It quantifies the log density of a data point within the distribution, which can be interpreted as a measure of its naturalness. The relative naturalness of a data point is determined by comparing its expectation w.r.t. its distribution. Definition (2) serves as a lower bound to Definition (1) by comparing the probability of a data point with the average probability of the distribution. Specifically, (1) can be rewritten as $e^{\log(p(x|\mathbf{pa}_X)) - [-H(X|\mathbf{pa}_X)]}$. The term $-\log(p(x|\mathbf{pa}_X))$ represents the measure of surprise in the information theory or say $\log(p(x))$ can indicate possibility that $x$ happens given $\mathbf{pa}_X$ and thus naturalness of a data point (Ash, 2012). The term $[-H(X|\mathbf{pa}_X)]$ represents the average naturalness of

---

[1] The term "naturalness" can be defined in various ways. In our current context, our definition of naturalness is exclusively determined by the information contained within Structural Causal Models (SCMs), rather than being influenced by the particular methods employed to carry out interventions. However, we are interested in investigating alternative definitions of naturalness in our future research.

the distribution. Therefore, $\log(p(x)) - [-H(X|\mathbf{pa}_X)]$ is the relative naturalness of a value $x$ given its parents compared to the average naturalness of the entire distribution $p(X|\mathbf{pa}_X)$. To normalize the average naturalness to 1, the final naturalness indicator is defined as $p(x|\mathbf{pa}_X)e^{H(X|\mathbf{pa}_X)}$, which is invariant to scaling transformations. For example, for a root endogenous variable following Gaussian distribution $\mathcal{N} \sim (X; \mu, \sigma^2)$, when $X = x$, its naturalness is $e^{\frac{\sigma^2 - (x-\mu)^2}{2\sigma^2}}$. Whatever the $\sigma$ is, the mean's naturalness is always $e^{\frac{1}{2}}$ and naturalness is reduces as data point is away from the mean. Following the same logic as (1), $p(x)$ could also indicate naturalness and thus, (2)'s rationality is justified.

(1) and (2)'s ideas directly depend on endogenous conditional distribution, while (3) and (4) go back to exogenous level. (3) and (4) is useful in practical implementations. For (3), to achieve SCMs in machine learning system, a common assumption is that the support of the distribution $p(X)$ does not contain disjoint sets, and a nonlinear, nonparametric generative model $X = f(\mathbf{pa}_X, U_X)$ is implemented, where $f$ is monotonically increasing w.r.t. $U_X$, which is usually assumed to be a standard Gaussian (Lu et al., 2020). In this case, under each local mechanism, noise distribution follows a standard Gaussian. Data points from tails can be thought of less impossible events. Similar to (4), following the same assumption, the CDF $P(x|\mathbf{pa}_X)$ and the CDF $P(U_X)$ is one-to-one mapping, i.e., specifically, $P(x|\mathbf{pa}_X) = P(u_X)$, where $x = f(\mathbf{pa}_X, u_X)$. For example, if $f$ is a linear SEM, assuming $U_X$ follows standard Gaussian distribution, $p(X|\mathbf{pa}_X) = \mathbf{N}(X; \mathbf{pa}_X, 1)$ and thus $P(X = x|\mathbf{pa}_X) = P(u_X)$, where $x = f(\mathbf{pa}_X, u_X)$. Hence, $P(X)$ can also represent naturalness.

These four definitions of naturalness provide different perspectives on measuring the naturalness of a data point based on its generative model. They capture various aspects such as the probability, entropy, relative naturalness, and the distribution of exogenous or endogenous variables.

### 4.2.2 NATURAL GENERATION

Accordingly, we define local $\epsilon$-natural generation as follows and introduces an epsilon threshold to determine whether a local mechanism is considered natural.

**Definition 2 (Local $\epsilon$-Natural Generation)** *Given a data point $X = x$, its parents taking a value, i.e., $\mathbf{PA}(X) = \mathbf{pa}_X$, and its local mechansim $p(X|\mathbf{PA}(X))$, according to Definition 2, the local mechanism $p(x|\mathbf{pa}_x)$ satisfies local $\epsilon$-natural generation, where $\epsilon$ is a small constant, if one of the following conditions is satisfied:*

*(1) $p(x|\mathbf{pa}_X)e^{H(X|\mathbf{pa}_X)} > \epsilon$: The naturalness of the data point, measured by $p(x)$ multiplied by the exponential of the entropy of the distribution, is greater than $\epsilon$.*

*(2) $\frac{p(x|\mathbf{pa}_X))}{\mathbb{E}[p(X|\mathbf{pa}_X)]} > \epsilon$: The naturalness of the data point, measured by the ratio of its probability to the average probability of the distribution, is greater than $\epsilon$.*

*(3) $\frac{\epsilon}{2} < P(u_X) < 1 - \frac{\epsilon}{2}$: The CDF of the exogenous variable $U_X$'s value $u_x$ of $x$ falls within the range $\left(\frac{\epsilon}{2}, 1 - \frac{\epsilon}{2}\right)$.*

*(4) $\frac{\epsilon}{2} < P(x|\mathbf{pa}_X) < 1 - \frac{\epsilon}{2}$: The CDF of $x$ given its parents' value $\mathbf{pa}_X$ falls within the range $\left(\frac{\epsilon}{2}, 1 - \frac{\epsilon}{2}\right)$.*

These criteria provide different perspectives on naturalness and allow for flexibility in assessing the naturalness of the generative process. The first two standards focus on the relative naturalness of the data point within the distribution, while the last two standards consider the positioning of the data point within the cumulative distribution function.

Note that definitions (3) and (4) focus on the tails of the distribution and aim to identify data points that deviate from the expected or typical values. In the case of a standard Gaussian distribution, the tails are considered less natural. These definitions provide a practical approach for detecting unnatural data points by leveraging the properties of the Gaussian distribution. Definition (3) is the most feasible and practical way to implement naturalness assessment, where we only need to deal with standard Gaussian distributions. It allows for the identification of data points in the tails, which are often considered less natural.

Finally, we define $\epsilon$-Natural Generation for $\mathbf{A}$-realization set $G(\mathbf{A})$ to judge whether all local mechanisms are natural. Here is the general definition:

**Definition 3 ($\epsilon$-Natural Generation)** *Given a SCM containing a set $\mathbf{X}$ taking value $\mathbf{x}$ and a set $G(\mathbf{X})$ containing $\mathbf{X}$ and its all endogenous ancestors, taking value $g(\mathbf{X})$, mechanisms of $G(\mathbf{X})$ satisfy $\epsilon$-natural generation, if $\forall \mathbf{N} \in G(\mathbf{X})$, each local mechanism $p(\mathbf{N} = \mathbf{n}|\mathbf{pa_N})$ satisfies local $\epsilon$-natural generation, where $\mathbf{n}$ and $\mathbf{pa_N}$ are values of $\mathbf{N}$ and its parents respectively, and $\epsilon$ is a small constant.*

### 4.3 MINIMAL CHANGE

**Minimal Change in Local Causal Mechanisms.** Recall that what we call local mechanisms are closely related to the distributions of the noise variables. Hence, we use the differences of the noise variable's CDFs in two worlds as the distance measures. Here we propose two distance measures in terms of a single local causal mechanism, called **local mechanism distance**, as follows :

**Theorem 4.1 (Distances Invariant to Distributions)** *For any two values $x_1, x_2$ associated with continous distribution $p(x)$, suppose the distance between the two points is given by $d(x_1, x_2) = ||P(x_1) - P(x_2)||_0$ or $d(x_1, x_2) = ||P(x_1) - P(x_2)||_1$, where $P(x^*) = \int_{-\infty}^{x^*} p(x)dx$. The two local mechanism distances are invariant to distribution, meaning that, when $p(x)$ is transformed to another continuous distribution $p(y)$ by reversible transformation, the distance of transformed points $x_1, x_2$ of $y_1, y_2$ is the same as that of $y_1, y_2$ before transformed.*

For example, for one endogenous node $\mathbf{V}_k \in G(\mathbf{A})$, its local mechanism distance is $d(\mathbf{u}_k, \mathbf{u}_k^*)$, where $\mathbf{u}_k$ is the value of $\mathbf{U}_k$, the noise variable of $\mathbf{V}_k$. since two proposed distances are invariant to different distributions, the distances of different noise variables are transformed into one space to be compared. If directly using the values of noise variables, the differences are hard to compared. For example, in linear SEMs, assuming $\mathbf{V}_1$ and $\mathbf{V}_2$ follows Gaussian distribution and uniform distribution respectively. The same value differences on $\mathbf{V}_1$ and $\mathbf{V}_2$ respectively is not comparable.

It is worth noting that when the $d$ takes $L^0$ or $L^1$ norm, it expresses different preferences on changes of mechanisms. Using $L^0$ norm tends to change the fewer number of nodes. Using $L^0$ norm prefers smaller total changes of mechanisms.

**Necessary Backtracking.** Causally prior variables should be changed as little as possible, in order to minimize the extent of backtracking. Moreover, some nodes may have more downstream nodes than others and should be changed less, as they will propagate changes to their descendants. Hence, we propose a notion of necessary backtracking. In the global mechanism distance, changes of each noise variables are independent and the distance on each single node has the same weight, which may sometimes lead to the bigger changes of causal earlier nodes. Hence, different weights should be set for different variables. Take global mechanism distance as an example. The modified distance is as follows:

$$D_M(g(\mathbf{A}), g(\mathbf{A})^*) = \sum_{\mathbf{u}_k \in \mathbf{u}_{g(\mathbf{A})}, \mathbf{u}_k^* \in \mathbf{u}_{g(\mathbf{A})^*}} w_k d(\mathbf{u}_k, \mathbf{u}_k^*) \tag{2}$$

$w_k$ is the weight of the node $U_k$, defined as the number of decedents of $\mathbf{V}_k$ denoted as $ND(\mathbf{V}_k)$. The number of variables influenced by a variable can be used as a weight in the distance measure. For example, in a causal graph where where $\mathbf{A}$ cause $\mathbf{B}$ and $\mathbf{C}$ is the confounder of $\mathbf{A}$ and $\mathbf{B}$. If $\mathbf{A} = \mathbf{a}^*$, $\mathbf{A}$'s and $\mathbf{B}$'s weight is 1 and 2 respectively.

### 4.4 A GENERAL FRAMEWORK FOR NATURAL COUNTERFACTUAL OPTIMIZATION

We can now formulate the problem of generating natural counterfactuals as an optimization problem. The objective is to minimize the distance of actual world and counterfactual world, i.e., the distance between $G(\mathbf{A})^*$ and $G(\mathbf{A})$, while satisfying the constraint of $\mathbf{A} = \mathbf{a}^*$ and $\epsilon$-natural generation of $G(\mathbf{A})^*$, where $\mathbf{A}^*$ is the counterfactual counterpart of $\mathbf{A}$. The optimization framework denoted as

Natural Counterfactual Optimization is defined as follows:

$$\min_{g(\mathbf{A})^*} D(g(\mathbf{A}), g(\mathbf{A})^*)$$
$$s.t. \quad \mathbf{A} = \mathbf{a}^* \tag{3}$$
$$s.t. \quad g(\mathbf{A})^* \text{ satisfying } \epsilon\text{-natural generation}$$

Where $g(\mathbf{A})$ is value of set $G(\mathbf{A})$ in the actual world, $g(\mathbf{A})^*$ is the counterfactual value to be optimized, $\epsilon$ is a small constant, and $D(\cdot)$ is a distance measure discussed above.

$g(\mathbf{A})^*$ may admit multiple optimal solutions and its solution set is denoted as $S_{g(\mathbf{A})}$. The distribution over $S_{g(\mathbf{A})}$ can be stipulated as:

$$p_n(G(\mathbf{A}) = g(\mathbf{A})^*|g(\mathbf{A}), change(\mathbf{A} = \mathbf{a}^*)) = \frac{p(g(\mathbf{A})^*)}{\int_{S_{g(\mathbf{A})}} p(G(\mathbf{A}))d(G(\mathbf{A}))} \tag{4}$$

The counterfactual world can be expressed as $< \mathcal{M}, p_n(G(\mathbf{A})^*|g(\mathbf{A}), change(\mathbf{A} = \mathbf{a}^*))p(\bar{\mathbf{U}}_{\mathbf{G}(\mathbf{A})}|\mathbf{E} = \mathbf{e}) >$, where $G(\mathbf{A})$ is sampled from $p_n(g(\mathbf{A})^*|g(\mathbf{A}), change(\mathbf{A} = \mathbf{a}^*))$ and values of noise variables $\bar{\mathbf{U}}_{\mathbf{G}(\mathbf{A})}$, which are the rest of noise nodes except noise variables of $G(\mathbf{A})$, is sampled from $p(\bar{\mathbf{U}}_{\mathbf{G}(\mathbf{A})}|\mathbf{E} = \mathbf{e})$.

## 5 CASE STUDIES

In this section, we put our theoretical framework into practice, examining its validity, utility, and performance through a series of empirical experiments on four simulated datasets and two publicly available datasets, MorphoMNIST and 3DIdentBOX. Please refer to the more detailed discussions in Sec. A.

On each experiment, we computed the average error between generated outcomes and ground-truth outcomes, which revealed a significant reduction in error when using our natural counterfactuals. This improvement stems from our approach's ability to perform necessary backtracking to determine plausible interventions when direct interventions are infeasible, while non-backtracking counterfactuals consistently employ direct interventions, even when they are implausible.

### 5.1 SIMULATION EXPERIMENTS

We start with a simulation dataset called *Toy* 1. We use the SCMs (shown in Sec. A.1) to generate 10000 data points as a training dataset and another 10000 data points as a test set. There are three endogenous variables $(n_1, n_2, n_3)$. $n_1$ is the confounder of $n_2$ and $n_3$ and $n_1$, and $n_2$ cause $n_3$.

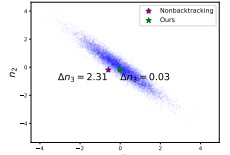 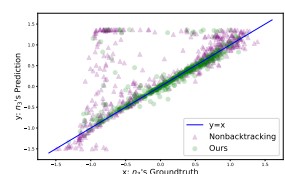

(a) Illustration of prediction error on a single sample

(b) Groundtruth-Prediction Scatter Plot

Figure 1: The Visualization Results on Toy 1.

**Experimental Settings.** Assume on Toy 1, only **data and qualitative causal graph** are known, **but not the ground-truth SCMs**. We employ normalizing flows to capture the causal mechanisms of variables $(n_1, n_2, n_3)$ through training on the provided training set, following Pawlowski et al. (2020b); Maaløe et al. (2019). **Given the learned SCMs and a data point from the test set as evidence, we do interventions with random values in non-backtracking counterfactuals or do changes in our natural counterfactuals either on $n_1$ or on $n_2$ within their support**. For our natural counterfactuals, given a small constant $\epsilon = 10^{-4}$, we use Eqn. 7 (refer to the concrete method in D of the Appendix) to learn feasible interventions, with $w_\epsilon = 10^4$ and the setting of other weights following Eqn. 7. We report the Mean Absolute Error (MAE) between our learned counterfactual outcomes and ground-truth outcomes on $n_2$ or/and $n_3$ on the test dataset repeated multiple times with multiple random seeds. Notice there may be no feasible interventions for some changes, as we have claimed, and thus we only report outcomes with feasible interventions, which is within the scope of our natural counterfactuals.

**Findings on Toy** 1. We first do *changes* or *interventions* on $n_2$. (1) **Prediction error comparison illustration on a single data point.** In Fig. 1, we evaluate the performance of counterfactual outcome estimation on a particular sample as an illustration. The green point represents the value of $(n_1, n_2)$ after doing change on $n_2$ in natural counterfactuals, the purple point indicates the value of $(n_1, n_2)$ after doing hard intervention on $n_2$ in non-backtracking counterfactuals, and the blue scatter plot shows the ground-truth support of $(n_1, n_2)$. We calculate the absolute error of $n_3$'s outcome given the value of the green point and the purple point, respectively, and find the error of the green point (with natural counterfactuals) is much smaller than that of the purple point (with non-backtracking counterfactuals), since the green point is a more feasible intervention compared with the purple one. In Fig. 1 (a), we also observe that the two points have the same value of $n_2$, while our natural counterfactuals use necessary backtracking to backtrack to $n_1$ and allow interventions on $n_1$. (2) **Prediction error comparison on whole Test Set.** As shown in Fig. 1 (b), a larger number of points with non-backtracking counterfactuals deviated from the line $y = x$, meaning that $n_3$'s prediction outcome is very different the ground-truth value. On the contrary, our outcomes are mainly located around the line $y = x$ (a small number of points may be far away from the line $y = x$ since the learned SCM is not perfect even within support). **Quantitative Results.** In Table 1, when putting *change* or *do* on $n_2$, Our MAE error is better than the non-backtracking by $61.6\%$, verifying the effectiveness of our approach.

We also do changes or interventions on $n_1$, where our natural counterfactuals will not do backtracking since $n_1$ is the root cause. Even in this setting, our result outperforms the non-backtracking, since our approach exclusive the points not satisfying -natural generation.

**Considering More Casual Graph Structures.** We also consider other causal graph structures and hence generate three more simulation datasets, i.e., Toy $2 - 4$. Toy $2 - 4$ contains 2, 4, and 3 nodes respectively, The results are reported in Table 1. More details are in Sec. A.1.

## 5.2 MORPHOMNIST

In this section, we investigate two types of counterfactuals using the MorphoMNIST dataset, comprising three variables: $(t, i, x)$. The causal graph, as presented in Fig. 2 (a) of the Appendix, indicates that $t$ (the thickness of the digit stroke) influences both $i$ (intensity of the digit stroke) and $x$ (images). Additionally, $i$ serves as the direct cause for $x$. A sample from this dataset is illustrated in Fig. 2 (b). The dataset encompasses 60,000 images in the training set and 10,000 in the test set.

Table 2: Ablation Study on $\epsilon$

| Model | $\epsilon$ | CFs | do(t) | | do(i) | |
|---|---|---|---|---|---|---|
| | | | $t$ | $i$ | $t$ | $i$ |
| V-SCM | - | NB | 0.336 | 4.532 | 0.283 | 6.556 |
| | $10^{-4}$ | | 0.314 | 4.506 | 0.171 | 4.424 |
| | $10^{-3}$ | Ours | 0.298 | 4.486 | 0.161 | 4.121 |
| | $10^{-2}$ | | 0.139 | 4.367 | 0.145 | 3.959 |
| H-SCM | - | NB | 0.280 | 2.562 | 0.202 | 3.345 |
| | $10^{-4}$ | | 0.260 | 2.495 | 0.105 | 2.211 |
| | $10^{-3}$ | Ours | 0.245 | 2.442 | 0.096 | 2.091 |
| | $10^{-2}$ | | 0.093 | 2.338 | 0.083 | 2.063 |

Our approach aligns with the experimental settings of the simulation experiments detailed in Sec. 5.1, with two notable exceptions. Firstly, for learning counterfactuals, we deploy two state-of-the-art deep learning models, specifically V-SCM (Pawlowski et al., 2020b) and H-SCM (Ribeiro et al., 2023). These models employ normalizing flows to discern causal relationships among the parent nodes of $x$, for instance, $(t, i)$ in the context of MorphoMNIST. Moreover, in determining $p(x|t, i)$, it's noteworthy that V-SCM incorporates VAE (Kingma & Welling, 2014) and HVAE (Maaløe et al., 2019). The second deviation pertains to our evaluation metric. Rather than utilizing MAE for outcome estimation, we adopt the counterfactual effectiveness metric as proposed by Ribeiro et al. (2023) and developed further by Monteiro et al. (2023). Once trained on the dataset, predictors for parent variables, given an $x$ value, can ascertain parent values such as $(t, i)$. Subsequently, we

Table 1: MAE Results on Toy $1 - 4$.

| Dataset | Toy 1 | | Toy 2 | Toy 3 | | | | | | Toy 4 | | |
|---|---|---|---|---|---|---|---|---|---|---|---|---|
| *do* or *change* | do($n_1$) | | do($n_2$) | do($n_2$) | do($n_1$) | | | do($n_2$) | | do($n_3$) | do($n_1$) | do($n_2$) |
| Outcome | $n_2$ | $n_3$ | $n_3$ | $n_3$ | $n_2$ | $n_3$ | $n_4$ | $n_3$ | $n_4$ | $n_4$ | $n_2$ | $n_3$ | $n_3$ |
| Non-backtracking | 0.477 | 0.382 | 0.297 | 0.315 | 0.488 | 0.472 | 0.436 | 0.488 | 0.230 | 0.179 | 0.166 | 0.446 | 0.429 |
| Ours | 0.434 | 0.354 | 0.114 | 0.303 | 0.443 | 0.451 | 0.423 | 0.127 | 0.136 | 0.137 | 0.158 | 0.443 | 0.327 |

Table 3: Results on Weak-3DIdent and Stong-3DIdent

| Dataset | Counterfactuals | $d$ | $h$ | $v$ | $\gamma$ | $\alpha$ | $\beta$ | $b$ |
|---------|-----------------|-----|-----|-----|----------|----------|---------|-----|
| Weak-3DIdent | Non-backtracking | 0.025 | 0.019 | 0.035 | 0.364 | 0.27 | 0.077 | 0.0042 |
| | Ours | 0.024 | 0.018 | 0.034 | 0.349 | 0.221 | 0.036 | 0.0041 |
| Stong-3DIdent | Non-backtracking | 0.100 | 0.083 | 0.075 | 0.387 | 0.495 | 0.338 | 0.0048 |
| | Ours | 0.058 | 0.047 | 0.050 | 0.298 | 0.316 | 0.139 | 0.0047 |

compute the absolute error between the parent values post either hard or feasible intervention and their predicted equivalents. This calculation is based on the images that the learned SCM produces when provided with $(t, i)$ inputs.

**Quantitative Results of** $change(i)$ **or** $do(i)$ **and Ablation Study on Naturalness Threshold** $\epsilon$**.** We employ two models, V-SCM and H-SCM, to execute counterfactuals with varying values of $\epsilon$ using $change(i)$ or $do(i)$. As depicted in Table 2, our error diminishes as $\epsilon$ increases when using the same inference model, because a higher $\epsilon$ opts for more feasible interventions. For instance, when considering H-SCM alongside $change(i)$ and $do(i)$, our method reduces errors from 48% at $\epsilon = 10^{-4}$ to 59% at $\epsilon = 10^{-2}$, compared to backtracking.

## 5.3 3DIDENTBOX

In this task, we utilize the 3DIdentBOX dataset from the collection cited in (Bizeul et al., 2023). Specifically, we focus on Weak-3DIdent and Strong-3DIdent. Both share the causal graph displayed in Fig. 4 (a) in Sec. A.3, with an image variable $x$ and its 7 parent variables. Variables $(d, h, v)$ determine the depth, horizontal, and vertical position of the teapot in image $x$. The angles of the teapot are controlled by $(\gamma, \alpha, \beta)$. Variable $b$ indicates the image's background color. Notably, Fig. 4 (a) reveals causal relationships among three parent variable pairs of $x$: $(h, d)$, $(v, \beta)$, and $(\alpha, \gamma)$. The distinction between Weak-3DIdent and Strong-3DIdent lies in the strength of these relationships. Weak-3DIdent presents milder associations as seen in Fig. 4 (b), whereas Strong-3DIdent shows stronger ones in Fig. 4 (c).

We adopt the exact same setting as in the MorphoMNIST experiments. Here, with $\epsilon = 10^{-3}$, we employ H-SCM as the inference model. We intervene or alter $(d, \beta, \gamma)$. The results are presented in Table 3. In both datasets, our approach outperforms the non-backtracking method. However, in Strong-3DIdent, our method has a larger margin over the non-backtracking approach since non-backtracking faces more unfeasible interventions when executing hard interventions using Strong-3DIdent. We also visualize results on Strong-3DIdent, and these are displayed in Fig. 5, confirming the effectiveness of our approach.

## 6 CONCLUSION AND DISCUSSION

This study presents novel concepts of naturalness criteria and minimal changes for generating practical natural counterfactuals. These prioritize feasible, data-backed interventions. By devising an optimization problem that minimizes world scenario distances while ensuring $\epsilon$-natural generation, we offer a structured approach to natural counterfactual reasoning. Our method's efficacy is underscored by diverse deep learning-based case studies. In the case of arbitrary machine learning systems that do not incorporate our causal graph, users have the option to rely on established background causal knowledge acquired through experiments, expert insights, or other authoritative sources. Alternatively, users can choose to employ data-driven methods to discover causal relationships, as long as these methods are implemented with appropriate assumptions and considerations taken into account. It is important to note that our approach significantly diverges from prior-based backtracking counterfactuals (von Kügelgen et al., 2022). In prior-based backtracking counterfactuals, interventions are limited to noise variables, potentially resulting in unnecessary alterations. Conversely, our approach selectively employs backtracking only when essential, aiming to minimize changes when direct intervention is unviable. For a more comprehensive understanding of these distinctions, please refer to Sec. C and Sec. B.

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

## A    MORE DETAILED DISCUSSION ON CASE STUDIES

In this section, we put our theoretical framework into practice, examining its validity, utility, and performance through a series of empirical experiments on four simulated datasets and two publicly available datasets, MorphoMNIST and 3DIdentBOX.

### A.1    SIMULATION EXPERIMENTS

We start with a simulation dataset called *Toy* 1. We use the following SCMs to generate 10000 data points as a training dataset and another 10000 data points as a test set:

$$n_1 = u_1\,, \qquad\qquad u_1 \sim \mathcal{N}(0,1)\,,$$
$$n_2 = -n_1 + \frac{1}{3}u_2\,, \qquad\qquad u_2 \sim \mathcal{N}(0,1)\,,$$
$$n_3 = \sin 0.25\pi(0.5n_2 + n_1) + 0.2u_3\,, \qquad u_3 \sim \mathcal{N}(0,1)\,,$$

where there are three endogenous variables $(n_1, n_2, n_3)$ and three noise variables $(u_1, u_2, u_3)$. $n_1$ is the confounder of $n_2$ and $n_3$ and $n_1$ and $n_2$ causes $n_3$.

**Experimental Settings.** Assume on Toy 1, only **data and qualitative causal graph** are known, **but not the ground-truth SCMs**. We employ **normalizing flows** to capture the causal mechanisms of variables $(n_1, n_2, n_3)$ through training on the provided training set. **Given the learned SCMs and a data point from the test set as evidence, we do *interventions* with random values in non-backtracking counterfactuals or do *changes* in our natural counterfactuals either on $n_1$ or on $n_2$ within their support**. For our natural counterfactuals, given a small constant $\epsilon = 10^{-4}$, we use Eqn. 7 to learn feasible interventions, with $w_\epsilon = 10^4$ and the setting of other weights following

Eqn. 7. We **report the Mean Absolute Error (MAE) between our learned counterfactual outcomes and ground-truth outcomes on $n_2$ or/and $n_3$ on the test dataset repeated multiple times with multiple random seeds.** Notice there may be no feasible interventions for some changes, as we have claimed, and thus we only report outcomes with feasible interventions, which is within the scope of our natural counterfactuals.

**Findings on Toy** 1. We first do *changes* or *interventions* on $n_2$. (1) **Prediction error comparison illustration on a single data point.** In Fig. 1, we evaluate the performance of counterfactual outcome estimation on a particular sample as an illustration. The green point represents the value of $(n_1, n_2)$ after doing change on $n_2$ in natural counterfactuals, the purple point indicates the value of $(n_1, n_2)$ after doing hard intervention on $n_2$ in non-backtracking counterfactuals, and the blue scatter plot shows the ground-truth support of $(n_1, n_2)$. We calculate the absolute error of $n_3$'s outcome given the value of the green point and the purple point, respectively, and find the error of the green point (with natural counterfactuals) is much smaller than that of the purple point (with non-backtracking counterfactuals), since the green point is a more feasible intervention compared with the purple one. In Fig. 1 (a), we also observe that the two points have the same value of $n_2$, while our natural counterfactuals use necessary backtracking to backtrack to $n_1$ and allow interventions on $n_1$. (2) **Prediction error comparison on whole Test Set.** As shown in Fig. 1 (b), a larger number of points with non-backtracking counterfactuals deviated from the line $y = x$, meaning that $n_3$'s prediction outcome is very different the ground-truth value. On the contrary, our outcomes are mainly located around the line $y = x$ (a small number of points may be far away from the line $y = x$ since the learned SCM is not perfect even within support). **Quantitative Results.** In Table 5, when putting *change* or *do* on $n_2$, Our MAE error is better than the non-backtracking by $61.6\%$, verifying the effectiveness of our approach.

We also do changes or interventions on $n_1$, where our natural counterfactuals will not do backtracking since $n_1$ is the root cause. Even in this setting, our result outperforms the non-backtracking, since our approach exclusive the points not satisfying -natural generation.

**Considering More Casual Graph Structures.** We also consider othrt causal graph structures and hence generate three more simulation datasets, i.e., Toy $2 - 4$. In Toy 2, there are two variables $(n_1, n_2)$ and $n_1$ causes $n_2$. As shown in Table 5, even in such a simple case, our approach still advances the non-backtracking when putting *do* or *change* on $n_1$. Toy 3 contains four variables, where $n_1$ is the confounder of $n_2$ and $n_4$ and $n_2$ causes $n3$ which is also the parent node of $n_4$. Three variables are in Toy 4 and they form a chain, i.e., $n_1$ causes $n_2$ and then $n3$. Similarly, in these experiments, ours also performs better.

## A.2    MORPHOMNIST

In this section, we study two types of counterfactuals on the dataset called MorphoMNIST, which contains three variables $(t, i, x)$. From the causal graph shown in Fig. 2 (a), $t$ (the thickness of digit stroke) is the cause of both $i$ (intensity of digit stroke) and $x$ (images) and $i$ is the direct cause of $x$. Fig. 2 (b) shows a sample from the dataset. The dataset contains 60000 images as the training set and 10000 as the test set.

We follow the experimental settings of simulation experiments in Sec. 5.1, except for two differences. One is that we use two state-of-the-art deep learning models, namely V-SCM (Pawlowski

(a) Illustration of prediction error on a single
sample

(b) Groudtruth-Prediction Scatter Plot

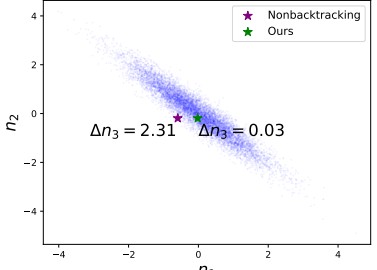
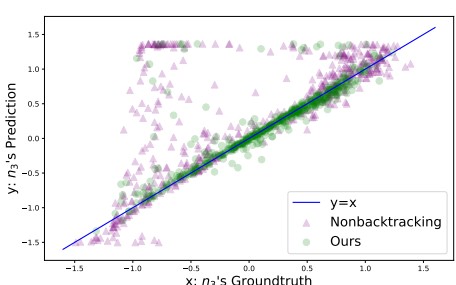

Table 4: The Visualization Results on Toy 1.

Table 5: MAE Results on Toy $1-4$.

| Dataset | | Toy 1 | | Toy 2 | | Toy 3 | | | | | Toy 4 | | |
|---|---|---|---|---|---|---|---|---|---|---|---|---|---|
| $do$ or $change$ | | $do(n_1)$ | | $do(n_2)$ | $do(n_2)$ | | $do(n_1)$ | | | $do(n_2)$ | $do(n_3)$ | $do(n_1)$ | $do(n_2)$ |
| Outcome | | $n_2$ | $n_3$ | $n_3$ | $n_3$ | $n_2$ | $n_3$ | $n_4$ | $n_3$ | $n_4$ | $n_4$ | $n_2$ $n_3$ | $n_3$ |
| Non-backtracking | | 0.477 | 0.382 | 0.297 | 0.315 | 0.488 | 0.472 | 0.436 | 0.488 | 0.230 | 0.179 | 0.166 0.446 | 0.429 |
| Ours | | 0.434 | 0.354 | 0.114 | 0.303 | 0.443 | 0.451 | 0.423 | 0.127 | 0.136 | 0.137 | 0.158 0.443 | 0.327 |

Table 6: MorphoMNIST results of $change(i)$ or $do(i)$ using V-SCM

| Intersection between Ours and NB | | | (NCO=1, NB=1) | (NCO=1, NB=0) | (NCO=0, NB=1) | (NCO=0, NB=0) |
|---|---|---|---|---|---|---|
| Number of Intersection | | | 5866 | 3135 | 0 | 999 |
| Non-backtracking | $t$'s MAE | 0.283 | 0.159 | 0.460 | 0 | 0.449 |
| | $i$'s MAE | 6.560 | 3.970 | 8.930 | 0 | 14.26 |
| Ours | $t$'s MAE | 0.161 | 0.150 | 0.181 | 0 | 0.461 |
| | $i$'s MAE | 4.121 | 3.825 | 4.675 | 0 | 14.16 |

et al., 2020b) and H-SCM (Ribeiro et al., 2023), as the backbones to learn counterfactuals. They use normalizing flows to learn causal relationships among $x$'s parent nodes, e.g., $(t, i)$ in MorphoM-NIST. Further, to learn $p(x|t, i)$, notice that V-SCM uses VAE (Kingma & Welling, 2014) and HVAE (Maaløe et al., 2019). Another difference is that, instead of estimating the outcome with MAE, we follow the same metric called counterfactual effectiveness in Ribeiro et al. (2023) developed by Monteiro et al. (2023), First, trained on the dataset, parent predictors given a value of $x$ can predict parent values, i.e., $(t, i)$'s, and then measure the absolute error between parent values after hard intervention or feasible intervention and their predicted values, which is measured on image the Learned SCM generates given the input of $(t, i)$.

**Quantitative Results of $change(i)$ or $do(i)$.** We use V-SCM to do counterfactual task of $change(i)$ (where $\epsilon = 10^{-3}$) or $do(i)$ with multiple random seeds on test set. In Table 6, the first column shows the MAE of $(t, i)$, indicating our results outperform that of non-backtracking. Next, we focus on the rest four-column results. In both types of counterfactuals, we use the same value $i$ in $do(i)$ and $change(i)$. Hence, after inference, we know which image satisfying $\epsilon$-natural generation in the two types of counterfactuals. In "NCO=1" of the table, NCO indicates the set of counter-

Table 7: Ablation Study on $\epsilon$

| Model | $\epsilon$ | CFs | do($t$) | | do($i$) | |
|---|---|---|---|---|---|---|
| | | | $t$ | $i$ | $t$ | $i$ |
| V-SCM | - | NB | 0.336 | 4.532 | 0.283 | 6.556 |
| | $10^{-4}$ | | 0.314 | 4.506 | 0.171 | 4.424 |
| | $10^{-3}$ | Ours | 0.298 | 4.486 | 0.161 | 4.121 |
| | $10^{-2}$ | | 0.139 | 4.367 | 0.145 | 3.959 |
| H-SCM | - | NB | 0.280 | 2.562 | 0.202 | 3.345 |
| | $10^{-4}$ | | 0.260 | 2.495 | 0.105 | 2.211 |
| | $10^{-3}$ | Ours | 0.245 | 2.442 | 0.096 | 2.091 |
| | $10^{-2}$ | | 0.093 | 2.338 | 0.083 | 2.063 |

factuals after natural counterfactual optimization. Notice that NCO set does not mean the results of natural counterfactuals, since some results do still not satisfy $\epsilon$-natural generation after natural counterfactual optimization. "NCO=1" mean the set containing data points satisfying $\epsilon$-natural generation and "NCO=0" contains data not satisfying $\epsilon$-natural generation after natural counterfactual optimization. Similarly, "NB=1" means the set containing data points satisfying naturalness criteria. (NCO=1, NB=1) presents the intersection of "NCO=1" and "NB=1". Similar logic is adopted to the other three combinations. The number of counterfactual data points are 10000 in two types of counterfactuals.

In (NCO=1, NB=1) containing 4135 data points, our performance is similar to the non-backtracking, showing natural counterfactual optimization tends to backtrack as less as possible when hard interventions have satisfied $\epsilon$-natural generation. In (NCO=1, NB=0), there are 3135 data points, which are "unnatural" points in non-backtracking counterfactuals. After natural counterfactual optimization, this huge amount of data points become "natural". In this set, our approach contributes to the maximal improvement compared to the other three sets in Table 6, improving 60.7% and 47.6% on thickness $t$ and intensity $i$. The number of points in (NCO=0, NB=1) is zero, showing the stability of our algorithm since our approach will not move the hard, feasible intervention into unfeasible intervention. Two types of counterfactuals perform similarly in the set (NCO=0, NB=0), also showing the stability of our approach.

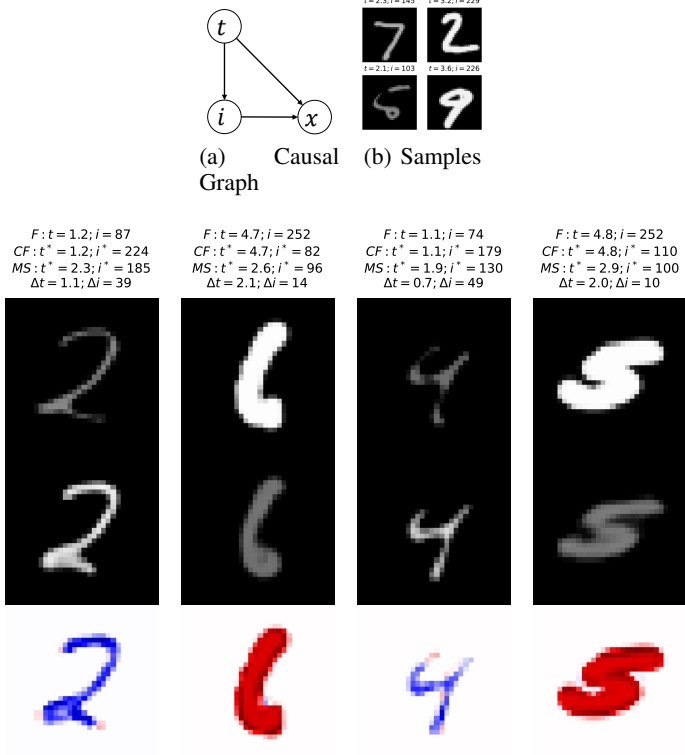

(a) Results of Non-backtracking Counterfactuals

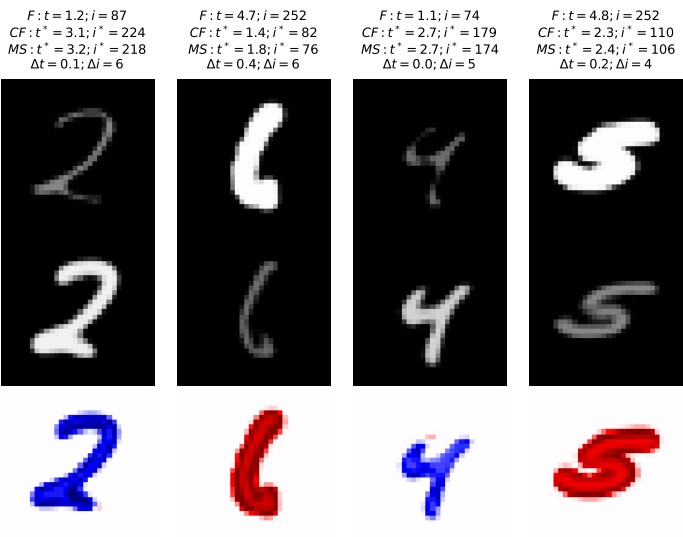

(b) Results of Natural Counterfactuals

Figure 3: Visualization Results on MorphoMNIST.

**Ablation Study on Naturalness Threshold $\epsilon$.** We use two models, V-SCM and H-SCM, to do counterfactuals with different values of $\epsilon$. As shown in Table 7, our error is reduced as the $\epsilon$ increases using the same inference model, since the higher $\epsilon$ will select more feasible interventions.

### A.3  3DIDENTBOX

In this task, we use more practical public datasets, 3DIdentBOX which contains multiple datasets (Bizeul et al., 2023). We use one of them, Weak-3DIdent and Strong-3DIdent. Their share the same

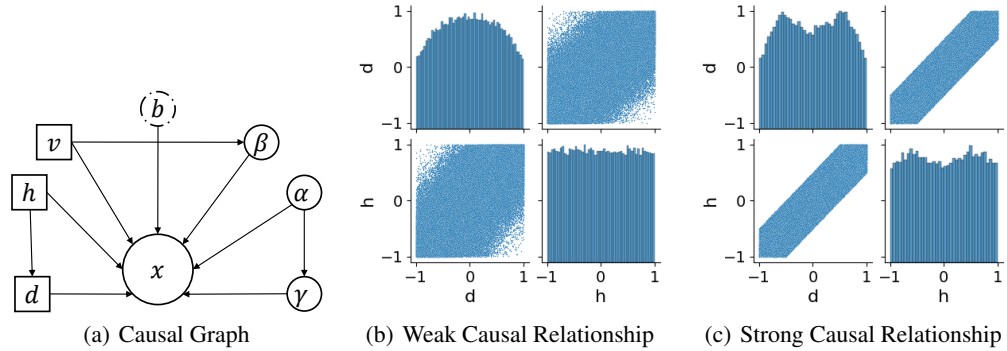

| (a) Causal Graph | (b) Weak Causal Relationship | (c) Strong Causal Relationship |

Figure 4: Causal graph of 3DIdent and the causal relationships of variables $(d, h)$ in Weak-3DIdent and Strong-3DIdent respectively.

Table 8: Results on Weak-3DIdent and Stong-3DIdent

| Dataset | Counterfactuals | $d$ | $h$ | $v$ | $\gamma$ | $\alpha$ | $\beta$ | $b$ |
|---|---|---|---|---|---|---|---|---|
| Weak-3DIdent | Non-backtracking | 0.025 | 0.019 | 0.035 | 0.364 | 0.27 | 0.077 | 0.0042 |
| | Ours | 0.024 | 0.018 | 0.034 | 0.349 | 0.221 | 0.036 | 0.0041 |
| Stong-3DIdent | Non-backtracking | 0.100 | 0.083 | 0.075 | 0.387 | 0.495 | 0.338 | 0.0048 |
| | Ours | 0.058 | 0.047 | 0.050 | 0.298 | 0.316 | 0.139 | 0.0047 |

causal graph as shown in Fig. 4 (a), containing one image variable $x$ and its 7 parent variables. Variables $(d, h, v)$ control the deepth, horizon position, and vertical position of the teapot of image $x$ respectively. Variables $(\gamma, \alpha, \beta)$ control three kinds angles of the teapot in an images. Variable $b$ represents background color of an image. As shown in Fig. 4 (a), there are causal relationship existing in three-pair parent variables of $x$, i.e., $(h, d)$, $(v, \beta)$ and $(\alpha, \gamma)$. There is one difference between Weak-3DIdent and Strong-3DIdent. In Weak-3DIdent, the variables of each pair exists weak causal relationship (Fig. 4 (b)), compared with that of Strong-3DIdent (Fig. 4 (c)).

We follow the exactly same setting as in the MophoMNIST experiments. Here, with $\epsilon = 10^{-3}$, we use H-SCM as inference model. We do intervention or change on $(d, \beta, \gamma)$. The results are shown in Table 8, in both dataset, our approach can perform better than the non-backtracking one but , in Strong-3DIdent, ours exists bigger margins over the non-backtracking method, since non-backtracking encounter more unfeasible interventions when doing hard interventions using Strong-3DIdent. We also do visualization on Strong-3DIdent. In Fig. 5, we show counterfactual outcomes in (a) and (b), where the text above each the first-row image (evidence) shows the error on the counterfactual outcome (the second-row corresponding image). Fig. 5 (a) shows counterfactual images (in the second row) which does not satisfy $\epsilon$-natural generation in the non-backtracking. (b) shows our results. Obviously, the images are even meaningless under the non-backtracking and they are had to recognized with bigger error. However, ours shows better visual effectiveness, showing that our solution could mitigate problems from hard interventions of the backtracking.

# B OBSERVATIONS ABOUT THE PRIOR-BASED BACKTRACKING COUNTERFACTUALS (VON KÜGELGEN ET AL., 2022)

## B.1 POSSIBILITY OF GRATUITOUS CHANGES

A theory of backtracking counterfactuals was recently proposed by von Kügelgen et al. (2022), which utilizes a prior distribution $p(\mathbf{U}, \mathbf{U}^*)$ to establish a connection between the actual model and the counterfactual model. This approach allows for the generation of counterfactual results under any condition by considering paths that backtrack to exogenous noises and measuring closeness in terms of noise terms. As a result, for any given values of $\mathbf{E} = \mathbf{e}$ and $\mathbf{A}^* = \mathbf{a}^*$, it is possible to find a sampled value $(\mathbf{U} = \mathbf{u}, \mathbf{U}^* = \mathbf{u}^*)$ from $p(\mathbf{U}, \mathbf{U}^*)$ such that $\mathbf{E}_{\mathcal{M}(\mathbf{u})} = \mathbf{e}$ and $\mathbf{A}^*_{\mathcal{M}^*(\mathbf{u}^*)} = \mathbf{a}^*$, as described in von Kügelgen et al. (2022). This holds true even in cases where $\mathbf{V} \setminus \mathbf{E} = \emptyset$ and $\mathbf{V}^* \setminus \mathbf{A}^* = \emptyset$, implying that any combination of endogenous values $\mathbf{E} = \mathbf{e}$ and $\mathbf{A}^* = \mathbf{a}^*$ can co-

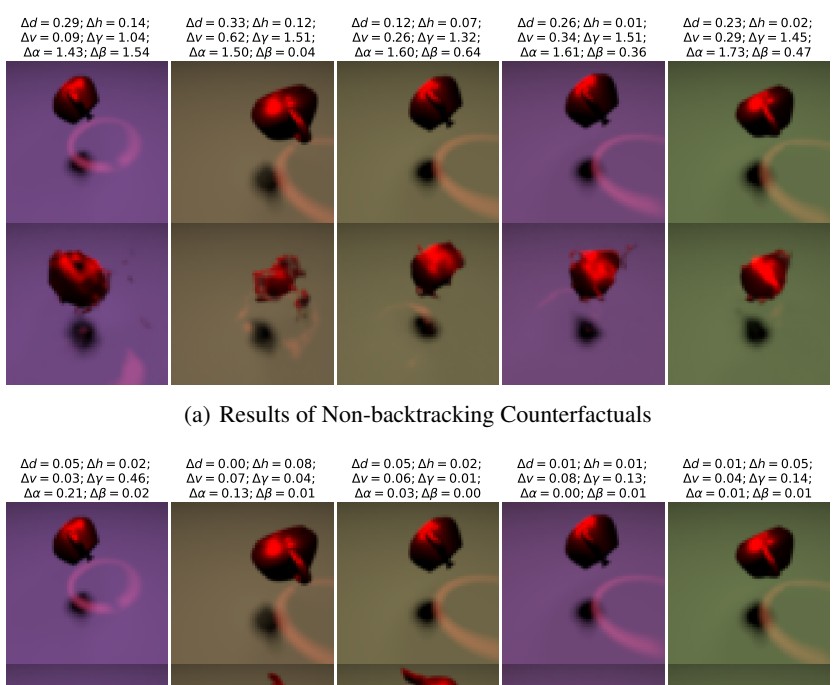

(a) Results of Non-backtracking Counterfactuals

(b) Results of Natural Counterfactuals

Figure 5: Visualization Results on Stong-3DIdent.

occur in the actual world and the counterfactual world, respectively. In essence, there always exists a path $(\mathbf{v} \rightarrow \mathbf{u} \rightarrow \mathbf{u}^* \rightarrow \mathbf{v}^*)$ that connects $\mathbf{V} = \mathbf{v}$ and $\mathbf{V}^* = \mathbf{v}^*$ through a value $(\mathbf{U} = \mathbf{u}, \mathbf{U}^* = \mathbf{u}^*)$, where $\mathbf{v}$ and $\mathbf{v}^*$ represent any values sampled from $p_{\mathcal{M}}(\mathbf{V})$ and $p_{\mathcal{M}^*}(\mathbf{V}^*)$, respectively.

However, thanks to this feature, this understanding of counterfactuals may allow for what appears to be gratuitous changes in realizing a counterfactual supposition. This occurs when there exists a value assignment $\mathbf{U}^* = \mathbf{u}^*$ that satisfies $\mathbf{E}^*_{\mathcal{M}^*(\mathbf{u}^*)} = \mathbf{e}$ and $\mathbf{A}^*_{\mathcal{M}^*(\mathbf{u}^*)} = \mathbf{a}^*$ in the same world. In such a case, intuitively we ought to expect that $\mathbf{E}^* = \mathbf{e}$ should be maintained in the counterfactual world (as in the factual one). However, there is in general a positive probability for $\mathbf{E}^* \neq \mathbf{e}$. This is due to the existence of at least one "path" from $\mathbf{E} = \mathbf{e}$ to any value $\mathbf{v}^*$ sampled from $p_{\mathcal{M}^*}(\mathbf{V}^*|\mathbf{A}^* = \mathbf{a}^*)$ by means of at least one value $(\mathbf{U} = \mathbf{u}, \mathbf{U}^* = \mathbf{u}^*)$, allowing $\mathbf{E}^*$ to take any value in the support of $p_{\mathcal{M}^*}(\mathbf{E}^*|\mathbf{A}^* = \mathbf{a}^*)$.

In the case where $\mathbf{A}^* = \emptyset$, an interesting observation is that $\mathbf{E}$ can take any value within the support of $p_{\mathcal{M}^*}(\mathbf{E}^*)$. Furthermore, when examining the updated exogenous distribution, we find that in Pearl's non-backtracking framework, it is given by $p_{\mathcal{M}^*}(\mathbf{U}^*|\mathbf{E}^* = \mathbf{e})$. However, in von Kügelgen et al. (2022)'s backtracking framework, the updated exogenous distribution becomes $p_B(\mathbf{U}^*|\mathbf{E} = \mathbf{e}) = \int p(\mathbf{U}^*|\mathbf{U})p_{\mathcal{M}}(\mathbf{U}|\mathbf{E} = \mathbf{e})d(\mathbf{U}) \neq p_{\mathcal{M}^*}(\mathbf{U}^*|\mathbf{E}^* = \mathbf{e})$, since using $\mathbf{u}^*$ sampled from $p(\mathbf{U}^*|\mathbf{U} = \mathbf{u})$ (where $\mathbf{u}$ is any value of $\mathbf{U}$) can result in any value of all endogenous variables $\mathbf{V}^*$. Therefore, von Kügelgen et al. (2022)'s backtracking counterfactual does not reduce to Pearl's counterfactual even when $\mathbf{A}^* = \emptyset$.

## B.2 ISSUES WITH THE DISTANCE MEASURE

In Equation 3.16 of von Kügelgen et al. (2022), Mahalanobis distance is used for real-valued $\mathbf{U} \in \mathbb{R}^m$, defined as $d(\mathbf{u}^*, \mathbf{u}) = \frac{1}{2}(\mathbf{u}^* - \mathbf{u})^{\mathrm{T}} \Sigma^{-1}(\mathbf{u}^* - \mathbf{u})$. However, it should be noted that the exogenous variables are not identifiable. There are several issues with using the Mahalanobis distance in this context.

Firstly, selecting different exogenous distributions would result in different distances. This lack of identifiability makes the distance measure sensitive to the choice of exogenous distributions.

Secondly, different noise variables may have different scales. By using the Mahalanobis distance, the variables with larger scales would dominate the distribution changes, which may not accurately reflect the changes in each variable fairly.

Thirdly, even if the Mahalanobis distance $d(\mathbf{u}^*, \mathbf{u})$ is very close to 0, it does not guarantee that the values of the endogenous variables are similar. This means that the Mahalanobis distance alone may not capture the similarity or dissimilarity of the endogenous variables adequately.

## C    DIFFERENCES BETWEEN NATURAL COUNTERFACTUALS AND NON-BACKTRACKING COUNTERFACTUALS (PEARL, 2009) OR PRIOR-BASED BACKTRACKING COUNTERFACTUALS (VON KÜGELGEN ET AL., 2022)

### C.1    DIFFERENCES BETWEEN NON-BACKTRACKING COUNTERFACTUALS AND OURS

Non-backtracking counterfactuals only do a direct intervention on target variable $A$, while our natural counterfactuals do backtracking when the direct intervention is implausible. Notice that when the direct intervention on $A$ is already plausible, our procedure of natural counterfactuals will be automatically distilled to the non-backtracking counterfactuals. In this sense, non-backtracking counterfactual reasoning is our special case.

### C.2    DIFFERENCES BETWEEN PRIOR-BASED BACKTRACKING COUNTERFACTUALS AND OURS

**(1) Intervention Approach and Resulting Changes:**

Prior-based Backtracking Counterfactuals: These counterfactuals directly intervene on noise/exogenous variables, which can lead to unnecessary changes in the counterfactual world. Consequently, the similarity between the actual data point and its counterfactual counterpart tends to be lower. In short, prior-based backtracking counterfactuals may introduce changes that are not needed.

Natural Counterfactuals: In contrast, our natural counterfactuals only engage in necessary backtracking when direct intervention is infeasible. This approach aims to ensure that the counterfactual world results from minimal alterations, maintaining a higher degree of fidelity to the actual world.

**(2) Counterfactual Worlds:**

Prior-based Backtracking Counterfactuals: This approach assigns varying weights to the numerous potential counterfactual worlds capable of effecting the desired change. The weight assigned to each world is directly proportional to its similarity to the actual world. it is worth noting that among this array of counterfactual worlds, some may exhibit minimal resemblance to the actual world, even when equipped with complete evidence, including the values of all endogenous variables. This divergence arises because by sampling from the posterior distribution of exogenous variables, even highly dissimilar worlds may still be drawn.

Natural Counterfactuals: In contrast, our natural counterfactuals prioritize the construction of counterfactual worlds that closely emulate the characteristics of the actual world through an optimization process. As a result, in most instances, one actual world corresponds to a single counterfactual world when employing natural counterfactuals with full evidence.

**(3) Implementation Practicality:**

Prior-based Backtracking Counterfactuals: The practical implementation of prior-based backtracking counterfactuals can be a daunting challenge. To date, we have been prevented from conducting a comparative experiment with this approach due to uncertainty about its feasibility in practical applications. Among other tasks, the computation of the posterior distribution of exogenous variables can be a computationally intensive endeavor. Furthermore, it is worth noting that the paper (von Kügelgen et al., 2022) provides only rudimentary examples without presenting a comprehensive algorithm or accompanying experimental results.

Natural Counterfactuals: In stark contrast, our natural counterfactuals have been meticulously designed with practicality at the forefront. We have developed a user-friendly algorithm that can be applied in real-world scenarios. Rigorous experimentation, involving four simulation datasets and two public datasets, has confirmed the efficacy and reliability of our approach. This extensive validation underscores the accessibility and utility of our algorithm for tackling specific problems, making it a valuable tool for practical applications.

## D  A CONCRETE METHOD FOR COUNTERFACTUAL GENERATION

In Sec. 4, we propose various theories for natural counterfactuals, including $\epsilon$-natural generation's definition, which $\mathbf{A}$-realization set $G(\mathbf{A})$ should follow when doing interventions on causally earlier variables, and tools for minimal changes containing various distance definitions to measure the difference between actual world and counterfactual world. We also provide how to do natural counterfactuals by the Natural Counterfactual Optimization (NCO) framwork. Here, we specify one method to apply to machine learning system, which can be as an example to help future researchers or practicers apply these high-level theories.

**Problem Setting.** Given data collected from real world and known causal graph, one can learn a SCM to model the data distribution. Given the full evidence $\mathbf{E} = \mathbf{e}$, we want to know, if there had been $\mathbf{A} = \mathbf{a}^*$, what would have happened to $\mathbf{B}$ using the learned SCM?

**A Specific Optimization Method.** Below is the equation of optimization:

$$\min_{g(\mathbf{A})^*} \sum_{\mathbf{v}_k \in g(\mathbf{A}), \mathbf{v}_k^* \in g(\mathbf{A})^*} w_k L^{(1)}(P(\mathbf{v}_k|\mathbf{pa}_i), P(\mathbf{v}_k^*|\mathbf{pa}_i^*))$$
$$s.t. \quad \mathbf{A} = \mathbf{a}^*$$
$$s.t. \quad \epsilon < P(\mathbf{v}_k^*|\mathbf{pa}_k^*) < 1 - \epsilon, \forall \mathbf{v}_k^* \in g(\mathbf{A})^* \tag{5}$$

Where $P(\cdot)$ means CDF, $w_k = ND(\mathbf{V}_k)$, $g(\mathbf{A})$ and $g(\mathbf{A})^*$ are actual and counterfactual values of $\mathbf{G}(\mathbf{A})$ respectively. (Please refer to other notions in Theorem 4). Here, we use the global mechanism distance as the measure of change and Definition (4) of local $\epsilon$-natural generation in Definition 2, since the definition is easy for be implementation. The CDF of the noise $\mathbf{u}_k$ has a one-to-one mapping with the conditional CDF of$\mathbf{v}_k$ given $\mathbf{pa}_k$, i.e., $P(\mathbf{v}_k|\mathbf{pa}_k) = P(\mathbf{u}_k)$ with $\mathbf{v}_k = f_k(\mathbf{pa}_k, \mathbf{u}_k)$. Hence, the Eqn. 6 becomes as below:

$$\min_{\mathbf{u}_{G(\mathbf{A})}^*} \sum_{\mathbf{u}_k \in \mathbf{u}_{G(\mathbf{A})}, \mathbf{u}_k^* \in \mathbf{u}_{G(\mathbf{A})}^*} w_k L^{(1)}(P(\mathbf{u}_k), P(\mathbf{u}_k^*))$$
$$s.t. \quad \mathbf{u}_\mathbf{A}^* = f_A^{-1}(\mathbf{a}^*, \mathbf{pa}_\mathbf{A}^*)$$
$$s.t. \quad \epsilon < P(\mathbf{u}_k^*) < 1 - \epsilon, \forall \mathbf{u}_k^* \in \mathbf{u}_k^* \in u_{G(\mathbf{A})}^* \tag{6}$$

Where the optimization parameter is changed from exogenous values into the values $\mathbf{u}_{G(\mathbf{A})}^*$ of noise $\mathbf{U}_{G(\mathbf{A})}^*$, which is corresponding variables of $G(\mathbf{A})$. For simplicity, we use $\mathbf{A}$ as subscript as indicator of terms related to $\mathbf{A}$, instead of number subscript. In practice, the Lagrangian method is used to optimize our objective function and the corresonding loss is as below:

$$\mathcal{L}(\mathbf{u}_{G(\mathbf{A})}^* \setminus \mathbf{u}_\mathbf{A}^*; \mathbf{u}_{G(\mathbf{A})}^*) = \sum_{\mathbf{u}_k \in \mathbf{u}_{G(\mathbf{A})}, \mathbf{u}_k^* \in \mathbf{u}_{G(\mathbf{A})}^*} w_k L^{(1)}(P(\mathbf{u}_k), P(\mathbf{u}_k^*))$$
$$+ w_\epsilon \sum_{\mathbf{u}_k^* \in \mathbf{u}_{G(\mathbf{A})}^*} \max((\epsilon - P(\mathbf{u}_k^*)), 0) + \max(\epsilon + P(\mathbf{u}_k^*) - 1, 0)) \tag{7}$$
$$s.t. \quad \mathbf{u}_\mathbf{A}^* = f_A^{-1}(\mathbf{a}^*, \mathbf{pa}_\mathbf{A}^*)$$

where the first term is the measure of distance between two world and the second term is the constraint of $\epsilon$-natural generation. where $w_\epsilon$ is a constant hyperparameter to punish noise values in the tails of noise distributions. Notice that $\mathbf{u}_\mathbf{A}^*$ is not optimized explicitly, since the value $\mathbf{pa}_\mathbf{A}^*$ is fully determined by other noise values $\mathbf{u}_{G(\mathbf{A})}^* \setminus \mathbf{u}_\mathbf{A}^*$ and $\mathbf{a}^*$ is an constant, with reversible function $f_A^{-1}$, $\mathbf{u}_\mathbf{A}^*$ is fully determined by $\mathbf{u}_{G(\mathbf{A})}^* \setminus \mathbf{u}_\mathbf{A}^*$ and $\mathbf{a}^*$.

In the next section, we use Eqn. 7 to do natural counterfactual optimization in multiple case studies of machine learning.

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
