# OpenReview forum: "Natural Counterfactuals With Necessary Backtracking"
_ICLR.cc/2024/Conference — Submitted to ICLR 2024_

### Official Review · Reviewer_WXgG · 2023-10-25

**Soundness:** 3 good
**Presentation:** 3 good
**Contribution:** 2 fair
**Rating:** 5
**Confidence:** 3

**Summary:**

The paper proposed a method called "natural counterfactuals" that is supposed to generate more plausible interventions compared to other methods such as the standard Pearlian approach.

**Strengths:**

- The proposed natural counterfactuals pose a novel and interesting approach compared to existing methods
- The paper is well written and structured

**Weaknesses:**

- The authors claim that their proposed method yields more plausible interventions. However, I can not see how this claim is evaluated in the case studies. Am I missing smth.?
- Knowledge about the causal graph looks like a strong assumption to me, but I know that this is assumption is quite common the causality domain. However, I miss a little bit the relation to machine learning -- i.e. how exactly this might be applied to arbitrary machine learning systems. This is somehow mentioned in Appendix D, but I was wondering whether it might be beneficial, in order to make the paper more accessible to a larger community, to elaborate more on this in the main paper.

Minor:
- Broken reference in Section 2 second line?
- Sometimes the authors write "non-backtracking" and sometimes "nonbacktracking" -- I am not a native speaker but the first one looks more correct to me
- V-SCM Reference in A.2 on page 12 is broken?

**Questions:**

See Weaknesses

---

> ### Author Response · Authors · 2023-11-18
> **Thank you for the constructive and encouraging comments**
>
> ***Q1. The authors claim that their proposed method yields more plausible interventions. However, I can not see how this claim is evaluated in the case studies. Am I missing smth.?***
>
> Thank you for your valuable feedback. In order to provide a clearer understanding of our case studies, we will commence by elaborating on why our approach excels at generating more realistic interventions in contrast to non-backtracking counterfactuals.
>
> **Theoretical Insight:** To gain a deeper understanding of how our approach operates, it is helpful to consider two distinct scenarios of direct interventions. First, when direct intervention on the target variable $A$ is infeasible, our approach aims to identify plausible interventions on causally antecedent variables by employing necessary backtracking. A pivotal requirement for any intervention in this context is its adherence to the naturalness criterion, which ensures that the resulting data points maintain a sufficient degree of naturalness. In the realm of machine learning, this is particularly advantageous, as machine learning models can more effectively handle data points that fall within the support of the training data distribution.
>
> Secondly, it is important to note that if a direct intervention on the target variable $A$ is already plausible, backtracking is not necessary, and our natural counterfactuals seamlessly fall back to the non-backtracking approach. This adaptability inherent in our method guarantees plausible interventions across a wide range of scenarios, resulting in counterfactual outcomes that consistently exhibit lower average errors compared to those generated by non-backtracking counterfactuals, which often but not always yield implausible interventions. Consequently, for the same set of evidence, our approach ensures that non-backtracking counterfactuals are only used when they do not rely on implausible interventions. Furthermore, when conducting counterfactual reasoning multiple times with the same evidence, our natural counterfactuals consistently ensure the plausibility of all interventions, whereas non-backtracking counterfactuals may falter in this regard.
>
> In summary, our approach consistently guarantees the plausibility of interventions, resulting in counterfactual outcomes characterized by lower average errors, in contrast to non-backtracking counterfactuals, which often require implausible interventions.
>
> **Experiments Demonstrating the Superiority of Our Approach in Generating Plausible Interventions:** In the main paper, we employ four methods to showcase that our counterfactual approach is capable of identifying more plausible interventions.
>
> **(1) Visualization of Counterfactual Performance on a Specific Sample (Fig. 1 (a)).**
>
> To visually illustrate the effectiveness of our counterfactuals in generating plausible interventions, we present Fig. 1 (a), which is constructed using the simulated dataset *Toy 1*. In this dataset, three endogenous variables, denoted as $(n_1, n_2, n_3)$, are present. $n_1$ serves as a confounder for $n_2$ and $n_3$, and both $n_1$ and $n_2$ influence $n_3$.
>
> We evaluate the performance of counterfactual outcome estimation on a specific sample for illustrative purposes. The green point in the figure represents the values of $(n_1, n_2)$ after performing a change on $n_2$, denoted as $change(n_2)$, using natural counterfactuals. In contrast, the purple point indicates the values of $(n_1, n_2)$ after applying a hard intervention on $n_2$, denoted as $do(n_2)$, in non-backtracking counterfactuals. The blue scatter plot represents the ground-truth support of $(n_1, n_2)$.
>
> **(1.1) Evaluating Plausiblity Using Probability Density.**
>
> Fig. 1 (a) offers valuable insights into understanding plausible intervention scenarios. As depicted in the figure, the purple point generated under non-backtracking counterfactuals is located outside the support of $(n_1, n_2)$. This indicates that a direct intervention as represented by the purple point is implausible and unlikely to occur in practice. In contrast, the natural counterfactual point (the green one) falls within the support of $(n_1, n_2)$. This is because our natural counterfactuals involve necessary backtracking to the variable $ n_2$, ensuring that the data point remains situated in an area with relatively high data density, as demonstrated in the figure. In other words, the data point adheres to our naturalness constraint.

---

> ### Author Response · Authors · 2023-11-18
>
> **(1.2) Evaluating Plausiblity Using Outcome Error.**
>
> Using plausible counterfactual input to predict outcomes usually leads to lower prediction error.
> Therefore, to quantitatively assess the performance, we calculate the absolute error of the outcome for $n_3$ based on the values represented by the green and purple points, respectively. We find that the error for the green point (generated using natural counterfactuals) is only $0.03$ which is significantly smaller than the error of $2.31$ for the purple point (generated using non-backtracking counterfactuals). This discrepancy in error underscores that the green point represents a more feasible (plausible) and realistic intervention compared to the purple one.
>
>
>
>
> **(2) Evaluating Plausiblity Using Outcome Error on Multiiple Samples (Fig. 1 (b)).**
>
> Similar to **(1.2)** above, we further visualize the performance of our counterfactual approach on multiple samples in Fig. 1 (b). Utilizing the simulation dataset *Toy 1*, the figure highlights significant differences between the outcomes generated by non-backtracking counterfactuals and our approach. In the figure, many points obtained through non-backtracking counterfactuals deviate from the line $y=x$, indicating substantial disparities between the predicted outcome for $n_3$ and the ground truth value. In contrast, the outcomes generated by our approach are predominantly clustered around the line $y=x$ (with a few exceptions, possibly due to the imperfections in the learned Structural Causal Model). This clustering effect is a result of our approach's ability to consistently ensure the plausibility of interventions in theory. This stark contrast in outcomes underscores the superiority of our method in comparison to non-backtracking counterfactuals, which can occasionally yield implausible interventions that lead to significant discrepancies from the expected outcomes.
>
>
> **(3) Evaluating Plausiblity Using Mean Absolute Error on Multiiple Samples Across Different Datasets.**
>
> To quantitatively assess the performance of our counterfactual approach, we conducted experiments on four simulation datasets and two publicly available datasets. In each dataset, we employed the same 10,000 samples as evidence to perform two types of counterfactual reasoning. Consequently, we obtained two sets of 10,000 counterfactual outcomes corresponding to non-backtracking counterfactuals and natural counterfactuals, respectively. We then calculated the Mean Absolute Error (MAE) for each set of 10,000 counterfactual outcomes. Across all these experiments, our natural counterfactuals consistently exhibited significantly lower errors compared to non-backtracking counterfactuals. For instance, when examining the intensity in MorphoMNIST, our method reduced the error by more than $39%$, illustrating its superior performance. This reduction in error is attributed to our method's ability to consistently ensure the plausibility of interventions in theory, a quality that non-backtracking counterfactual reasoning fails to achieve.
>
> **(4) Number of Generated Plausible Interventions.**
>
> Referring to Table 6 in the Appendix, particularly in the context of the MorphoMNIST dataset, our natural counterfactual approach exhibits a noteworthy advantage. When conducting counterfactual reasoning on the same set of 10,000 samples, our method generates $53.4%$ more plausible interventions in comparison to non-backtracking counterfactuals.
> This substantial improvement stems from our method's capability to transform an initially implausible direct intervention into a plausible one through necessary backtracking. Furthermore, it is worth noting that our approach encompasses all the plausible interventions identified in non-backtracking counterfactuals. This inclusivity arises from the fact that when a direct intervention on the target variable, such as intensity $i$ in MorphoMNIST, is already deemed plausible, our natural counterfactual procedure seamlessly aligns with non-backtracking counterfactuals, ensuring comprehensive coverage of plausible scenarios.
> For a more comprehensive understanding of these findings, please consult Table 6 and its detailed explanation in Section A.2 of the Appendix.
>
> In response to your feedback, we have introduced a paragraph at the outset of Section 5, Case Studies, to elucidate why our method excels in generating more plausible interventions across simulated and public datasets.

---

> ### Author Response · Authors · 2023-11-18
>
> **Q2. Knowledge about the causal graph looks like a strong assumption to me, but I know that this is assumption is quite common the causality domain. However, I miss a little bit the relation to machine learning -- i.e. how exactly this might be applied to arbitrary machine learning systems. This is somehow mentioned in Appendix D, but I was wondering whether it might be beneficial, in order to make the paper more accessible to a larger community, to elaborate more on this in the main paper.**
>
> Thanks for the helpful suggestion. Your critical point is well taken. It is certainly a limitation of our work and most of the related work that the causal information needed for such methods is not always available. Still, we believe counterfactual reasoning fundamentally relies on causal understanding. If machine learning aims also to harness counterfactual reasoning, it has to somehow incorporate causal knowledge or assumptions, either by building in background knowledge or by automated causal discovery. In this connection, it is worth noting that learning of causal structures from observational data, though still fraught with challenges, has made a number of impressive advances in recent decades.
>
> To acknowledge this important critical point, we have incorporated a discussion within Sec. 6 (Conclusion and Discussion) of the main paper. Thanks again for the suggestion.
>
>
> ***Q3. Minor Issues.***
>
> We are very grateful for your careful review of our paper. We have addressed and resolved all the problems you spotted, and your valuable feedback is highly appreciated.

---

> ### Author Response · Authors · 2023-11-22
> **Looking Forward to Your Additional Feedback**
>
> We would like to express our sincere gratitude for your time and effort in reviewing our work. Your insights and feedback have been invaluable to us.
>
> As the decision timeline for ICLR is nearing, we hope our response has clarified your initial concerns and questions. We would be happy to provide any further clarifications if necessary.
> We deeply appreciate your time and efforts and look forward to any additional feedback you may have.

---

### Official Review · Reviewer_CJwc · 2023-10-31

**Soundness:** 3 good
**Presentation:** 4 excellent
**Contribution:** 3 good
**Rating:** 6
**Confidence:** 1

**Summary:**

The paper proposes a new framework called "natural counterfactuals" for generating counterfactual explanations, aimed at overcoming limitations in Judea Pearl's traditional approach.  The new methodology modifies the standard nonbacktracking requirement in counterfactual reasoning. It permits alterations in variables that are causally prior to the target variables in a counterfactual scenario, but only when these changes are necessary to meet a "naturalness" criterion. To balance this flexibility, an optimization framework is introduced to minimize the extent of such backtracking. Through experiments, the paper demonstrates that this new approach is more effective in generating practical, realistic counterfactual explanations compared to the standard Pearlian method.

**Strengths:**

The paper provides a very clear rationale for why traditional "nonbacktracking" counterfactuals can be impractical, laying out the context and need for an alternative approach. The introduction of "natural counterfactuals" is a new take on counterfactual reasoning, addressing limitations in the standard Pearlian framework.  One of the primary goals of the paper is to produce actionable insights which would have direct implications. They formulate the problem of generating natural counterfactuals as a simple optimization problem.
The paper provides empirical results based on both simulated and real-world data, thereby demonstrating the efficacy and applicability of their methodology.

**Weaknesses:**

Due to my limited expertise in this area of research, I have not delved deeply into the core aspects of the paper. My question is high-level.
How does your work differ from 'Backtracking Counterfactuals'? https://arxiv.org/abs/2211.00472".

**Questions:**

NA

---

> ### Author Response · Authors · 2023-11-18
> **Thank you for the constructive and encouraging comments**
>
> ***Question: How does your work differ from 'Backtracking Counterfactuals'? https://arxiv.org/abs/2211.00472".***
>
> Thank you for this important question. Due to the space constraint, we did not discuss this question in detail in the main text but left it to Sec. C of the Appendix. In our paper, we term the "Backtracking Counterfactuals" in your question as **prior-based backtracking counterfactuals**. In addition to the discussion in Sec. C, we also raised some concerns regarding prior-based backtracking counterfactuals in Sec. B of the Appendix.
>
> We outline below the key differences between our natural counterfactuals and prior-based backtracking counterfactuals:
>
> **Prior-based Backtracking Counterfactuals** [1]:
> This approach involves backtracking to all exogenous variables, which amounts to requiring that interventions are applied only to these variables. The measure of similarity between the actual and counterfactual worlds is based on a predefined prior distribution. Exogenous variable values in the counterfactual world are sampled from a posterior distribution given the actual world's values and the desired counterfactual value.
>
> **Natural Counterfactuals**:
> Our method involves backtracking with a 'naturalness constraint' when an intervention on the target variable $A$ is implausible. This process aims to minimize changes and backtracking. It backtracks only when it is necessary to maintain "naturalness". If a direct intervention on $A$ is plausible, our approach aligns with non-backtracking counterfactuals.
>
>
> **Differences Between Them:**
>
> **(1) Intervention Approach and Resulting Changes:**
>
> Prior-based Backtracking Counterfactuals: These counterfactuals directly intervene on noise/exogenous variables, which can lead to unnecessary changes in the counterfactual world. Consequently, the similarity between the actual data point and its counterfactual counterpart tends to be lower. In short, prior-based backtracking counterfactuals may introduce changes that are not needed.
>
> Natural Counterfactuals: In contrast, our natural counterfactuals only engage in necessary backtracking when direct intervention is infeasible. This approach aims to ensure that the counterfactual world results from minimal alterations, maintaining a higher degree of fidelity to the actual world.
>
> **(2) Counterfactual Worlds:**
>
> Prior-based Backtracking Counterfactuals: This approach assigns varying weights to the numerous potential counterfactual worlds capable of effecting the desired change. The weight assigned to each world is directly proportional to its similarity to the actual world. it is worth noting that among this array of counterfactual worlds, some may exhibit minimal resemblance to the actual world, even when equipped with complete evidence, including the values of all endogenous variables. This divergence arises because by sampling from the posterior distribution of exogenous variables, even highly dissimilar worlds may still be drawn.
>
> Natural Counterfactuals: In contrast, our natural counterfactuals prioritize the construction of counterfactual worlds that closely emulate the characteristics of the actual world through an optimization process. As a result, in most instances, one actual world corresponds to a single counterfactual world when employing natural counterfactuals with full evidence.
>
> **(3) Implementation Practicality:**
>
> Prior-based Backtracking Counterfactuals:
> The practical implementation of prior-based backtracking counterfactuals can be a daunting challenge. To date, we have been prevented from conducting a comparative experiment with this approach due to uncertainty about its feasibility in practical applications. Among other tasks, the computation of the posterior distribution of exogenous variables can be a computationally intensive endeavor. Furthermore, it is worth noting that the paper [1] provides only rudimentary examples without presenting a comprehensive algorithm or accompanying experimental results.
>
> Natural Counterfactuals: In stark contrast, our natural counterfactuals have been meticulously designed with practicality at the forefront. We have developed a user-friendly algorithm that can be applied in real-world scenarios. Rigorous experimentation, involving four simulation datasets and two public datasets, has confirmed the efficacy and reliability of our approach. This extensive validation underscores the accessibility and utility of our algorithm for tackling specific problems, making it a valuable tool for practical applications.
>
> In response to your feedback, we have incorporated a discussion on these key differences into Sec. 6 (Conclusion and Discussion) of the main paper (highlighted in red) and expanded the explanation in Sec. C of the Appendix, covering all three types of counterfactual reasoning discussed in our paper.
>
> [1] Julius von Kugelgen, Abdirisak Mohamed, and Sander Beckers. Backtracking counterfactuals. arXiv preprint arXiv:2211.00472, 2022.

---

> ### Author Response · Authors · 2023-11-22
> **Looking Forward to Your Additional Feedback**
>
> We would like to express our sincere gratitude for your time and effort in reviewing our work. Your insights and feedback have been invaluable to us.
>
> As the decision timeline for ICLR is nearing, we hope our response has clarified your initial concerns and questions. We would be happy to provide any further clarifications if necessary.
> We deeply appreciate your time and efforts and look forward to any additional feedback you may have.

---

> > ### Comment · Reviewer_CJwc · 2023-11-23
> >
> > Thank you for clearly differentiating your work from Backtracking Counterfactuals and emphasizing this in your conclusion. I keep my score on the accept side.

---

### Official Review · Reviewer_adRY · 2023-11-05

**Soundness:** 2 fair
**Presentation:** 3 good
**Contribution:** 3 good
**Rating:** 5
**Confidence:** 3

**Summary:**

The authors provide an alternative framework for reasoning about counterfactuals that is claimed to provide more natural results than the prevailing framework.

**Strengths:**

The paper is well-written.

**Weaknesses:**

The example cited in the introduction does not seem to differentiate between: (1) a reasonable approach to reasoning about an "actual cause" (an event that is a cause of the outcome in this specific case); and (2) what the authors refer to as "constructive" reasoning (identifying plausible alternative events that could have changed the outcome). These are different goals, and both are useful. The former would be useful to identify causal chains, while the second is useful to determine what, in that causal chain, would have been a reasonable point of intervention to change the outcome.  This doesn't invalidate the results of the paper, but it casts the results in a different light. The approach outlined in the paper is not a replacement for the existing theory, but instead is an elaboration of it to allow for a wider range of types of reasoning.

It seems doubtful that "naturalness" can be defined in a way that is unambiguous and will be universally agreed upon. The specific "mechanisms" of intervention are generally external to the SCM. That is, the *way that an intervention changes the model* can be represented in an SCM (e.g., an atomic intervention substitutes for the structural function of the variable intervened upon), but the *manner in which that intervention achieves that result* is not represented in the SCM. Thus, the SCM itself does not appear to contain the information necessary to decide "naturalness". For example, the event "Tom is restrained from hitting Jerry" may be unlikely (unnatural) in a bus, but it would be entirely natural in a car. Specifically, suppose that Tom and Jerry are riding in a car, Tom is sitting the in the back seat, and Jerry is in the front seat. Tom could strike Jerry in the event of a sudden stop, but he wouldn't if he was wearing a safety belt.  Yes, it is possible that "Wearing a seat belt" could be represented in the SCM, but not all such "intervention variables" would be. This is a larger issue about "what SCMs are good for." They cannot represent everything, so what should we expect them to be able to reason about? The authors propose that "naturalness" is one of those things, but this increases the number of things that the variables in a given SCM should represent.

The paper would be improved by a more intuitive explanation of the simulation experiments and why they should provide readers with confidence that the approach outlined by the authors is valid and useful. It seems counter-intuitive that such a claim could be supported by simulation and the current text does little to explain the underlying logic.

The experiments in Section 5 seem more like demonstrations and less like useful tests that, if the proposed approach did not work well, would have shown that. That is, they are not particular severe tests of the proposed approach.

**Questions:**

What is the underlying logic behind the experiments in Section 5?

Why do they provide convincing evidence that we should expect the proposed methods to work well in nearly all cases?

---

> ### Author Response · Authors · 2023-11-18
> **Thank you for the constructive and encouraging comments**
>
> ***Q1. The example cited in the introduction does not seem to differentiate between: (1) a reasonable approach to reasoning about an "actual cause" (an event that is a cause of the outcome in this special case); and (2) what the authors refer to as "constructive" reasoning (identifying plausible alternative events that could have changed the outcome).***
>
> Thank you for your comment. We originally designed the example to highlight the importance of identifying plausible interventions when doing counterfactual reasoning. However, upon reflection, we recognize that certain phrasings used in the example might be misleading, implying an intention to identify and reason about "actual causes." In response, we have rephrased the short description of the bus example, highlighted in red.
>
> To clarify, our approach does not involve picking out an actual cause for an outcome. Instead, our focus lies in reasoning about and generating an alternative scenario with a specified difference from the actual. Further, considering the challenges associated with directly intervening on target variables, our objective is to identify plausible interventions through the novel concept of necessary backtracking. This approach advocates for making minimal but natural revisions according to one conception of "naturalness."

---

> ### Author Response · Authors · 2023-11-18
>
> ***Q2. It seems doubtful that "naturalness" can be defined in a way that is unambiguous and will be universally agreed upon. The specific "mechanisms" of intervention are generally external to the SCM. That is, the way that an intervention changes the model can be represented in an SCM (e.g., an atomic intervention substitutes for the structural function of the variable intervened upon), but the manner in which that intervention achieves that result is not represented in the SCM. Thus, the SCM itself does not appear to contain the information necessary to decide "naturalness". For example, the event "Tom is restrained from hitting Jerry" may be unlikely (unnatural) in a bus, but it would be entirely natural in a car. Specifically, suppose that Tom and Jerry are riding in a car, Tom is sitting the in the back seat, and Jerry is in the front seat. Tom could strike Jerry in the event of a sudden stop, but he wouldn't if he was wearing a safety belt. Yes, it is possible that "Wearing a seat belt" could be represented in the SCM, but not all such "intervention variables" would be. This is a larger issue about "what SCMs are good for." They cannot represent everything, so what should we expect them to be able to reason about? The authors propose that "naturalness" is one of those things, but this increases the number of things that the variables in a given SCM should represent.***
>
> We appreciate your insight on alternative conceptions of "naturalness," and we agree that different perspectives or contexts can favor different definitions. In our paper, we employ a specific notion of naturalness that we argue is useful for mitigating a serious problem in counterfactual generation, but we do not claim that this is the only sensible notion. To address your concern, we have included a footnote in the paper (Sec. 4.2) to clarify that our definition of naturalness is just one among several possible interpretations, and we are open to exploring other potential definitions of naturalness in future research.
>
> The specific conception of naturalness in our paper was chosen in part because measures of it can be defined based on information provided by a Structural Causal Model (SCM). This conception does not depend on the manner in which interventions are implemented or realized, or on other things that are not implied by a SCM. Your point that SCMs are limited in their representational capability is well taken, and we agree that the fact that our notion of naturalness is measured against a given SCM indicates certain limitations of this notion as well. However, this notion is useful in our task, or so we try to demonstrate in this paper, and gives rise to a principled framework for counterfactual generation that proves valuable in practical scenarios. In particular, it is valuable if the goal is to generate instances that accurately realize the desired counterfactual feature. By aligning with our naturalness criterion, we try to ensure that the generated data points do not appear as outliers within the overall data distribution (here we understand outliers as those that do not follow regular causal mechanisms). In other words, within our framework, "naturalness" is utilized to describe a specific property of generated data points, and an intervention is deemed a "plausible intervention" when the generated point adheres to the naturalness criterion associated with that intervention. Simultaneously, it presents an alternative perspective on the example you have given. The causal mechanism at work in a car can vary significantly from those in a bus, involving distinct variables and functions. Consequently, when we analyze the bus scenario within the confines of our framework, the causal mechanism that is suitable for a car may not be directly applicable to the bus scenario.
>
> Given that our notion of naturalness is measured against a SCM, our current approach depends essentially on the SCM machinery. We agree that the framework has various limitations, which will need to be addressed if alternative and richer conceptions of naturalness are invoked. At the same time, we hope our notion of naturalness is able to enrich the understanding of counterfactuals and improve their practical implications, at least in certain circumstances.

---

> ### Author Response · Authors · 2023-11-18
>
> ***Q3. What is the underlying logic behind the experiments in Section 5? Why do they provide convincing evidence that we should expect the proposed methods to work well in nearly all cases?***
>
> Thank you for your valuable questions and feedback. Our experiments serve to illustrate the effectiveness of natural counterfactuals in identifying plausible interventions and subsequently reducing errors in generating counterfactual outcomes when compared to non-backtracking counterfactuals. Our approach consistently ensures plausible interventions, keeping generated data points within the support of the data distribution. In contrast, non-backtracking counterfactuals do not provide such a guarantee.
>
> **Theoretical Insight:** Our work challenges the standard interpretation under which a counterfactual supposition is to be realized by directly intervening on a target variable, denoted as $A$, even when such an intervention is likely to be implausible according to the data distribution. Non-backtracking counterfactuals often require such implausible interventions, resulting in data points that deviate from the data distribution. In machine learning, these implausible interventions lead to significant deviations from the desired counterfactual feature, as machine learning models struggle to generalize to data points lying outside the training data distribution.
>
> In contrast, our method employs natural counterfactuals, especially when direct interventions on the target variable $A$ are unfeasible. We advocate for a backtracking approach that aims for minimal change while avoiding unnecessary modifications. Crucially, any intervention must adhere to a naturalness criterion, ensuring that the resulting data points remain plausible and realistic. In machine learning, this approach enables machine learning models to handle these data points effectively since they fall within the support of the training data distribution.
>
> It is worth noting that if a direct intervention on the target variable $A$ is already plausible, our natural counterfactuals align with the non-backtracking approach. This adaptability ensures plausible interventions across various scenarios, resulting in counterfactual outcomes with consistently lower average errors compared to those from non-backtracking counterfactuals, which may include implausible interventions.
>
> **Experiments in Machine Learning:** We conducted experiments using four simulation datasets and two public datasets, performing counterfactual analyses under both non-backtracking counterfactuals and our natural counterfactuals. We computed the average error between generated outcomes and ground-truth outcomes regarding the desired counterfactual feature, which revealed a significant reduction in error when using our natural counterfactuals. For example, our natural counterfactuals demonstrated more than a 30% reduction in error with the MorphoMNIST dataset, as shown in Table 2 of the main paper. This pattern held across other experiments involving four simulation datasets and one additional public dataset, affirming the superior accuracy of our natural counterfactuals. This improvement stems from our approach's ability to perform necessary backtracking to determine plausible interventions when direct interventions are infeasible, while non-backtracking counterfactuals consistently employ direct interventions, even when they are implausible.
>
> In response to your insightful feedback, we have added an introductory paragraph to Sec. 5, to better articulate the theoretical advantages of our method over non-backtracking counterfactuals and to elucidate the logic behind our experiments. We hope this clarification will make the reasons for the superior performance of our approach more transparent, allowing users to better understand its practical applications and requirements.

---

> ### Author Response · Authors · 2023-11-22
> **Looking Forward to Your Additional Feedback**
>
> We would like to express our sincere gratitude for your time and effort in reviewing our work. Your insights and feedback have been invaluable to us.
>
> As the decision timeline for ICLR is nearing, we hope our response has clarified your initial concerns and questions. We would be happy to provide any further clarifications if necessary.
> We deeply appreciate your time and efforts and look forward to any additional feedback you may have.

---

### Author Response · Authors · 2023-11-18
**Response to all the reviewers and area chairs**

We sincerely thank all the reviewers for their valuable comments. We are pleased to learn that our paper was found to be ``"well-written"``/``"well written and structured"`` and provided a ``"very clear rationale for why traditional "non-backtracking" counterfactuals can be impractical"`` (adRY, WXgG, CJwc). Additionally, we are grateful for the recognition of our approach as ``"new"``/``"novel and interesting"`` (CJwc, WXgG).
The acknowledgment of our empirical results demonstrating ``"the efficacy and applicability of their methodology"`` (CJwc) is particularly encouraging.

We would also highly appreciate it if reviewers could generously participate in the discussion to comment on our response. *Your continued feedback is invaluable to us, and we eagerly look forward to it.*

---

### Meta-Review · Area_Chair_ci1f · 2023-12-06

**Metareview:**

The reviewers attribute some merit the contribution and the topic (backtracking/interventional counterfactuals) is really interesting, however, my overall reading of the reviews is that the paper has still quite a lot of areas for improvement, particularly in terms of motivation and evaluation, which my own independent reading of the paper confirms. The rebuttal by the authors did not really cleared out the concerns expressed by the reviewers. As a result, I am unable to recommend acceptance but I encourage the authors to revise their paper in light of the reviews and resubmit to another top tier venue.

**Justification For Why Not Higher Score:**

None of the reviewers is strongly in favor of accepting the paper and they highlight a number of points for improvement.

**Justification For Why Not Lower Score:**

N/A

---

### Decision · Program_Chairs · 2024-01-16

Reject